# Dissociation of the nuclear basket triggers chromosome loss in aging yeast

Mihailo Mirkovic[1], Jordan McCarthy[1], Anne Cornelis Meinema[1], Julie Parenteau[2], Sung Sik Lee[1,3], Sherif Abou Elela[2], Yves Barral[1]*

[1]Department of Biology, Institute of Biochemistry, ETH Zürich, Zürich, Switzerland; [2]Département de microbiologie et d'infectiologie, Faculté de médecine et des sciences de la santé, Université de Sherbrooke, Sherbrooke, Canada; [3]Scientific Center for Optical and Electron Microscopy, ETH Zürich, Zürich, Switzerland

## eLife Assessment

This **fundamental** study reveals that aging in yeast leads to chromosome mis-segregation due to asymmetric partitioning of chromosomes, driven by disruption of the nuclear pore complex and pre-mRNA leakage. The findings are **convincingly** supported by carefully-designed experimental data with a combination of genetic, molecular biology and cell biology approaches.

*For correspondence:
yves.barral@bc.biol.ethz.ch

Competing interest: The authors declare that no competing interests exist.

**Abstract** In many organisms, aging is a clear risk factor for chromosome missegregation, the main source of aneuploidy. Here, we report that old yeast cells lose chromosomes by partitioning them asymmetrically to their daughter cells together with the pre-existing (old) spindle pole body (SPB, centrosome equivalent in yeast). Strikingly, remodelling of the nuclear pore complex (NPC) and the displacement of its nuclear basket triggered these asymmetric chromosome segregation events. Simultaneously, nuclear basket displacement caused unspliced pre-mRNAs to leak into the cytoplasm. We show that removing the introns of three genes involved in chromosome segregation was sufficient to fully suppress chromosome loss in old cells. Promoting pre-mRNA leakage in young cells also caused asymmetric chromosome partitioning and loss through the same three introns. Therefore, we propose that basket displacement from NPCs and its consequences for pre-mRNA quality control are key triggers of aging phenotypes such as aneuploidy.

## Introduction

Cellular aging results in conserved phenotypes, including chromosome mis-segregation, fragmentation of mitochondria, accumulation of damaged or misfolded proteins, and alteration of nuclear morphology (*Denoth Lippuner et al., 2014*; *López-Otín et al., 2023*; *Janssens and Veenhoff, 2016*). Importantly, the NPC is remodelled in aging across evolutionary lineages, as shown in fungi, worms, and mammals (*Rempel et al., 2020*; *Meinema et al., 2022*; *Rempel et al., 2019*). As the primary gateway between the nucleus and cytoplasm, the NPC controls mRNA export and protein distribution between these two cellular compartments. The most prominent change in the NPCs of old cells is the loss of their nuclear basket, which lies on the nucleoplasmic side of the NPC. Nuclear basket loss is observed both in dividing cells and upon expression of the progeric variant of Lamin A, progerin (*Jin et al., 2019*; *Morlot et al., 2019*). The protein TPR (Mlp1 and Mlp2 in *S. cerevisiae*) is the main structural component of the nuclear basket in mammals. The nuclear basket serves as a critical processing hub for mRNA export and nuclear retention of pre-mRNA, protein quality control, and gene regulation (*Møller et al., 2018*; *Bitterman et al., 2003*; *Denoth-Lippuner et al., 2014*; *Shcheprova et al., 2008*; *Baldi et al., 2017*; *Boveri, 2008*). Altering the structure or function of the NPC or its nuclear

basket could have diverse and wide-ranging effects on cellular function, given the NPC's broad range of functions and central position in eukaryotic cellular information flow. Despite this and the fact that the reorganization of NPC architecture with age is prominent in a wide range of organisms, little is known about whether and how NPC remodelling contributes to aging.

The unicellular fungus *Saccharomyces cerevisiae*, budding yeast, is a remarkably useful system for studying the role of NPCs in cellular aging (*Denoth Lippuner et al., 2014*; *Janssens and Veenhoff, 2016*). Budding yeast undergoes replicative aging, during which the mother cell produces a finite number of daughter cells before dying (*Jacobs et al., 1961*). After the mother cell enters senescence, the rate of cell division slows. Importantly, several studies have emphasized that the yeast NPC is remodelled during replicative aging (*Rempel et al., 2020*; *Meinema et al., 2022*; *Rempel et al., 2019*), but the exact consequences of its reorganization are not fully understood.

Extrachromosomal DNA circles (also called eccDNAs) accumulate in the mother cell nucleus and are one of the key drivers of aging in the budding yeast (*Sinclair and Guarente, 1997*; *Jin et al., 2019*; *Morlot et al., 2019*). These DNA circles form during the cell's lifetime through excision of genomic segments, generally by recombination between repeated sequences (*Møller et al., 2018*). The vast majority of these circles (>99%) stems from the highly repetitive rDNA locus and are referred to as extrachromosomal rDNA circles (ERCs) (*Sinclair and Guarente, 1997*). As the cell ages, these circles accumulate exponentially because they are replicated in S-phase and retained in the mother cell at mitosis (*Sinclair and Guarente, 1997*; *Bitterman et al., 2003*). By the end of their life, yeast cells contain about 1000 ERCs, which represents roughly the same amount of DNA as the rest of the genome (*Bitterman et al., 2003*). Importantly, their retention in the mother cell relies on their anchorage to NPCs via the acetyltransferase complex SAGA (*Meinema et al., 2022*; *Denoth-Lippuner et al., 2014*). These NPC-ERC units are then confined to the mother part of the dividing nucleus by a lateral diffusion barrier forming in the nuclear envelope in the future plane of cleavage (*Denoth-Lippuner et al., 2014*; *Shcheprova et al., 2008*; *Baldi et al., 2017*). Yeast cells undergo a closed mitosis, meaning that they keep their nuclear envelope intact throughout mitosis. Remarkably, anchorage of ERCs to NPCs results in the dissociation of their nuclear basket, which is a direct result of SAGA-driven acetylation of nucleoporins on the nucleoplasmic side of the NPC, including Nup60 (*Denoth-Lippuner et al., 2014*). Consequently, most of the NPCs of old yeast cells lack a nuclear basket (*Meinema et al., 2022*). However, it is unknown whether nuclear basket displacement has consequences for the physiology and survival of the cell or contributes to the aging process.

One way to approach the question of whether nuclear basket displacement from the NPCs contributes to the aging process is to determine whether other aging phenotypes depend on basket displacement and if so, how. In this study, we focused on chromosome instability in old yeast cells as a case study of an aging phenotype. In many metazoans, including humans, aging results in an increasing frequency of chromosome missegregation and aneuploidy, correlating with increasing risk of cancer (*Boveri, 2008*; *Jacobs et al., 1961*). Our understanding of the molecular mechanisms of chromosome segregation errors in old cells remains rudimentary. Here, we report evidence that NPC remodelling during yeast aging leads to chromosome loss from old mother cells and characterize the underlying mechanisms.

## Results

### Aging yeast cells lose chromosomes

To examine whether aging affects the karyotype of yeast cells, we visualized and followed the fate of individual chromosomes in replicatively aging mother cells, using either TetO-labeled chromosome II or chromosome IV as reporters. Chromosome IV is the second-longest chromosome, while chromosome II is of average size. Both chromosomes were labeled with an array of 256 TetO repeats in the vicinity of their centromere (*Neurohr et al., 2011*; *Kruitwagen et al., 2018*). These chromosomes were visualized by expressing fluorescently tagged versions of the TetR protein, which binds to TetO repeats (TetR-GFP or TetR-mCherry; *Figure 1A–B*). We then monitored budding yeast mother cells throughout their entire replicative lifespan using live-imaging microscopy combined with a microfluidics platform (*Meinema et al., 2022*; *Jo et al., 2015*). The replicative age of each mother cell was measured by counting the number of daughters that it budded off over time, defining how many budding events they have already completed at a given time point (completed budding events, CBE).

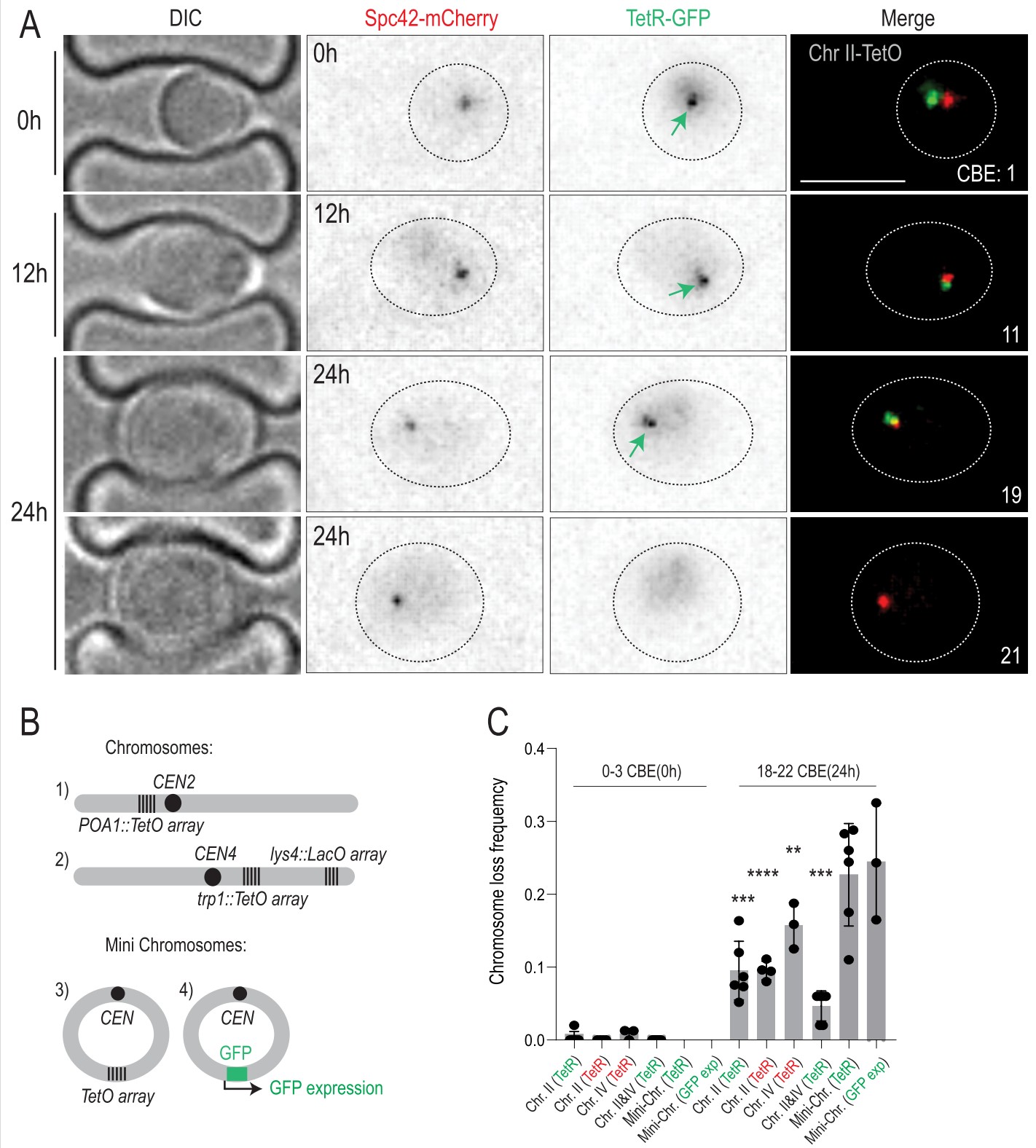

**Figure 1.** Old cells lose chromosomes. (**A**) Cells carrying labeled chromosome II (Chr II) and imaged at different ages on the microfluidic chip. Time of acquisition is indicated on the left. Fluorescent markers are indicated on top. Replicative age of imaged cells (completed budding event – CBE) is indicated in merged images. Scale bar (upper right panel) is 5 µm. Green arrow marks the TetR-GFP foci. (**B**) Schematic representation of the chromosome and minichromosome labels used in this study. (**C**) Chromosome loss frequency of indicated chromosomes after 0 hr (~0–3 CBE) and

*Figure 1 continued on next page*

*Figure 1 continued*

24 hr (~18–22 CBE) of imaging. The mode of chromosome visualization is indicated in parenthesis, i.e., TetR-GFP (green), TetR-mCherry (red), or mini-chromosome encoded GFP (GFP exp). Chromosome loss is defined as the absence of the label in the mother cell in G1 shortly after the final cell cycle, or the final anaphase. Each data point represents frequency of chromosome loss in a cohort of ~50 cells. Unpaired t-test (*<0.05, **<0.005, ***<0.0005). Mean value ± SD.

The online version of this article includes the following source data and figure supplement(s) for figure 1:

**Source data 1.** Complete sequences of plasmids PYB 546 and PYB 2665.

**Figure supplement 1.** Introns affect chromosome loss, but not whole nuclei missegregation in aging.

Bud size and the number and position of the spindle pole bodies (SPBs, labeled with mCherry) were used to identify the cell cycle stages at which the cells were imaged (*Figure 1A*).

Analysis of these images showed that all young mother cells (~0–3 CBE; t=0 hr) and their daughters contained a fluorescent dot, documenting the presence of the labeled chromosomes in all of them. However, when reaching the age of 18–22 CBE (t=24 hr), 10–15% of these now old mother cells had lost their dot (*Figure 1B–C*). This was observed irrespective of which chromosome was labeled, and which fluorescent tag was used. Fluorescence of the labeled SPB was not affected, indicating that dot loss was not caused by fluorophore instability in old cells. A chromosome dot could be lost from aging mother cells in three possible ways: (1) Chromosome missegregation during anaphase (2) Migration of the entire nucleus into the daughter cell (3) Through the loss of a TetO array. The SPB is located in the nuclear envelope, so the presence of the SPB but the absence of the chromosome points to chromosome loss during anaphase, rather than the missegregation of the entire nucleus into the daughter cell (*Figure 1A*). Nuclear missegregation into the daughter cells was previously reported during replicative aging and is marked by the segregation of both SPBs to the daughter compartment (*Figure 1—figure supplement 1A–B*; *Crane et al., 2019*). While we do observe whole nuclei missegregation in ~3% of the cells, this is a distinct event from chromosome loss, which occurs in ~10–15% of final replicative divisions.

To test if the loss of fluorescent chromosomal foci was due to the loss of the TetO array specifically or of the entire chromosome, we included a second label (an array of 256 repeats of the LacO sequence) in the middle of the long arm of chromosome IV (*Figure 1B*) and asked whether the cells that lost the TetO array lost the LacO repeats as well. Analysis of these images showed that 85% of the cells lacking the TetO array had indeed lost the LacO array (50/59). Thus, these cells had probably lost the entire chromosome. Supporting this notion, loss of the TetO array was fatal to the cells. More than 99% (145/146 chromosome II loss events) of the dot-less cells failed to divide further, as expected for a haploid cell losing an entire chromosome. Thus, chromosome loss took place specifically in the last division of old mother cells and likely caused their death.

Finally, we examined whether the loss of a reporter chromosome points to general chromosome instability in old cells. To assess this possibility, we quantified how often aging cells simultaneously lost both chromosome II and chromosome IV (*Figure 1B–C*). As expected, the co-loss rate (~5%) was lower than the individual loss rates (~10–15%), but notably higher than the ~1% expected by chance alone. Therefore, the loss of a single chromosome does not necessarily accompany the loss of another, but the high rate of simultaneous missegregation points to general chromosome instability in old cells. Considering that the budding yeast haploid genome has 16 chromosomes, the high loss frequency of a single chromosome and the fact that individual chromosomes are lost in not fully independent manners, we estimate that 35–70% of aging mother cells likely lose several chromosomes in their final division, whereas the other cells die with an intact karyotype. We concluded that, much like many other organisms, yeast undergoes a burst of chromosome instability in old age (*Figure 1A and C*).

In contrast to chromosomes, extra-chromosomal DNA circles accumulate in the yeast mother cell with age rather than being lost (*Sinclair and Guarente, 1997*; *Denoth-Lippuner et al., 2014*). Similarly, deleting the centromere from a chromosome causes the retention of both sister chromatids in the mother cell (*Kruitwagen et al., 2018*). Thus, we wondered whether centromeres drive chromosome loss in old mother cells. DNA circles containing a centromeric sequence (*CEN*) were lost from aged mother cells at a similar rate to full-size chromosomes (~20% of old cells; *Figure 1B–C*). Removing the centromere of the exact same circle was shown in previous work to cause its retention in the mother cell rather than its loss (*Meinema et al., 2022*; *Denoth-Lippuner et al., 2014*; *Shcheprova et al., 2008*). Therefore, centromeres are necessary and sufficient for chromosomes and mini

chromosomes to be lost from aging mother cells. To exclude the possibility that the TetO array used to label them is the cause of chromosome and mini-chromosome loss in aging, we also characterized the stability of a circular mini-chromosome expressing short-lived GFP as a marker instead of TetO repeats. Scoring of the GFP signal demonstrated that old mother cells lost these mini-chromosomes at essentially the same frequency as they lost the TetO-labeled one (*Figure 1B–C*). We concluded that centromeres drive chromosome loss in old cells.

## Aged cells asymmetrically partition both sister chromatids to the bud

Given the prominent role of the centromere in chromosome loss, we rationalized that the loss of chromosomes in old mother cells could be due to mitotic missegregation events. To determine whether it was the case, we next focused our analysis on cells imaged during mitosis in time-lapse recordings of aging cells (*Figure 2A*). Analysis of anaphases in old mother cells revealed a striking increase in the frequency of chromosome missegregation, using labeled chromosome II, IV, and minichromosome as reporters (*Figure 2B*). In 600 total anaphase events captured in our movies, we found that 66 (11%) of them co-segregated the labeled sister chromatids to a single pole with 54/66 (82%) of the missegregation events resulting in asymmetric partition of the chromatids to the bud of these old mother cells (*Figure 2C*). This was virtually never observed in young cells (*Figure 2A–B*). Remarkably, the frequency of chromosome loss in the old mother cells and of sister chromatids segregating asymmetrically to the bud were in the same size order, irrespective of whether chromosome II or chromosome IV was labeled. Therefore, chromosome loss from old mother cells seems fully explained by a rise in sister chromatid non-disjunction with age, resulting in co-segregation of the sister chromatids asymmetrically to the bud. Next, we examined the mechanism behind this peculiar mode of directed chromosome missegregation.

In budding yeast, the pre-existing (old) SPB, which the cell inherited from the previous mitosis, segregates in a highly biased manner to the bud (~95% of mitoses) (*Lengefeld et al., 2017*; *Pereira et al., 2001*; *Hotz et al., 2012*). Consequently, labeling of stable SPB components, such as the core SPB protein Spc42, with mCherry causes the pre-existing SPB to shine bright and segregate in 95% of mitoses to the bud (*Figure 2A*, red arrowhead), while the new one is dim and remains in accordance with proportions in the mother cell. This useful effect is a result of the folding kinetics of mCherry. In short, mCherry matures slowly (~45 min half-time) relative to the duration of yeast mitosis (~50 min from spindle assembly to completion of cytokinesis), and therefore the new mCherry-labeled SPB is dimmer than the old one. Given the mother-daughter bias of SPB inheritance and sister chromatid missegregation with age, we wondered whether the non-disjoining sister chromatids of old cells attach to and co-segregate with the old SPB. Thus, we examined the partition of the labeled chromosomes relative to the pre-existing and new SPBs in old mother cells (t=24 hr, *Figure 2D and E*). In about 10% of old cells, SPB inheritance is inverted, meaning that the old SPB goes to the mother cell instead of the daughter (*Lengefeld et al., 2017*; *Pereira et al., 2001*). In a strong majority of these cases (80%), the non-disjoining sister chromatids followed the old SPB to the mother cell, instead of going to the bud (*Figure 2D and E*). We conclude that non-disjoining sister chromatids of old mother cells remained attached to the pre-existing SPB and partitioned with it to the bud, causing them to lose these chromosomes. In other words, SPB identity mattered more than mother-daughter identity in determining which cell inherited the mis-segregated chromosomes.

In budding yeast, sister chromatids both initially attach to the pre-existing SPB, which is available first. These syntelic attachments are then dissolved by the aurora B kinase, allowing sister chromatids to reorient, reach bi-orientation, and partition symmetrically in mitosis (*Biggins et al., 1999*; *Cheeseman et al., 2002*). Failure to correct these initial attachment errors causes sister chromatids to co-segregate with the old SPB to the bud (*Tanaka et al., 2002*). The similarity between this phenotype and the mechanism of chromosome loss in old mother cells suggested, therefore, that the error correction pathway might be impaired in old yeast cells. To test this possibility, we analysed the effects of dampening the function of aurora B (called Ipl1 in budding yeast) during aging. Young cells carrying the temperature-sensitive *ipl1-321* mutant allele and grown at a permissive temperature (27 °C) showed no detectable chromosome loss (*Figure 2F*). However, they started losing chromosomes early in age (*Figure 2F*), resulting in a shortened lifespan (*Figure 2—figure supplement 1A*). Strikingly, the chromosome loss frequency of the old *ipl1-321* mutant cells (t=24 h) was indistinguishable from that of the wild-type cells, indicating that the effects of age and reducing Ipl1 activity were

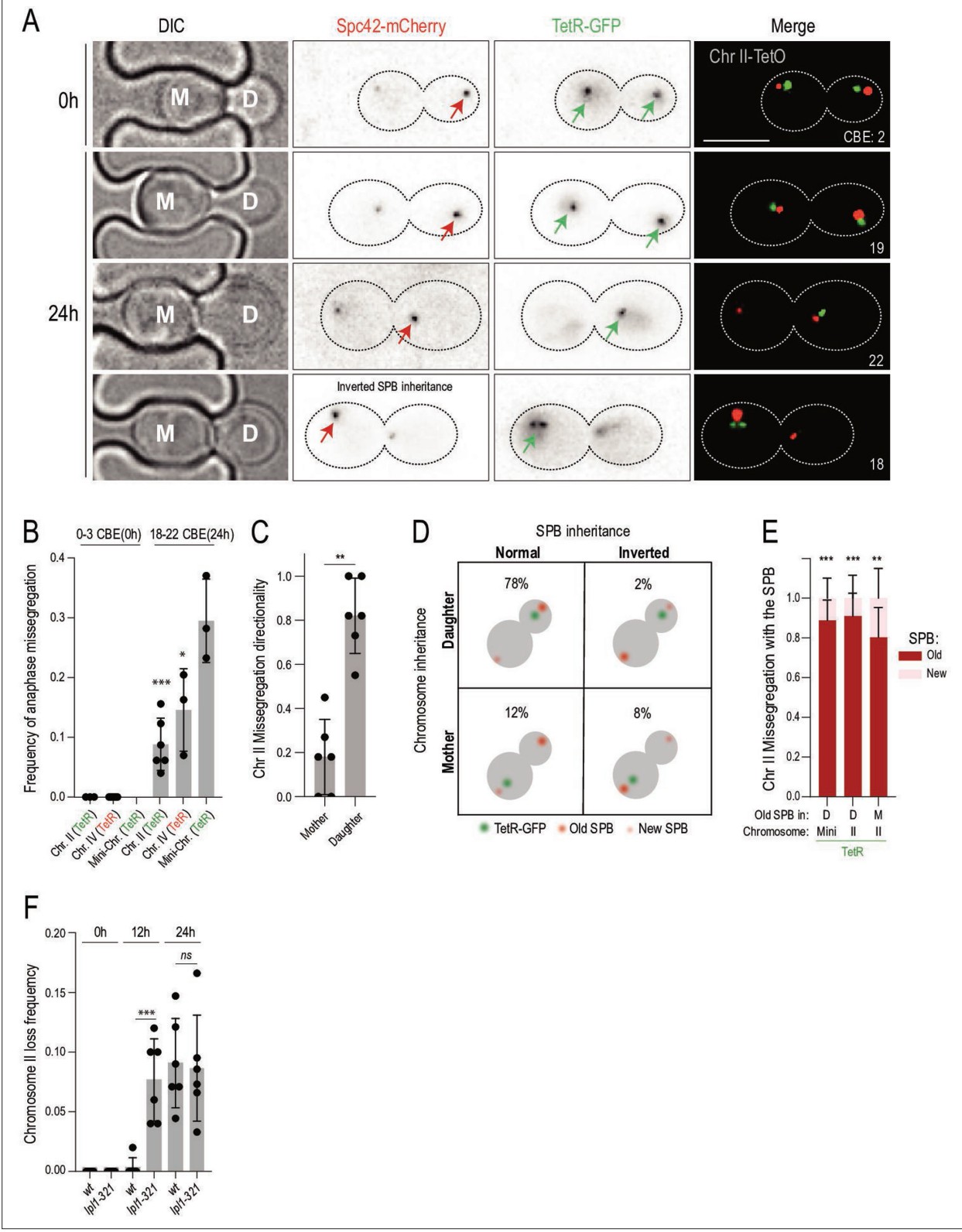

**Figure 2.** Old cells lose chromosomes by asymmetric sister chromatid partitioning. (**A**) Images of anaphase cells carrying the labeled chromosome II as in *Figure 1A*. Red arrows mark the old spindle pole bodies (SPBs).Scale bar (upper right panel) is 5 µm. (**B**) Frequency of anaphase missegregation events where chromosomes are visible only in the mother or daughter cell (n>150) Bars are labeled as in *Figure 1C*. Each data point represents frequency of anaphase missegregation in the cohort of ~50 anaphases. Mean value ± SD. (**C**) Fraction of anaphase missegregation events that lead to

*Figure 2 continued on next page*

*Figure 2 continued*

TetR-GFP foci being present only in mother or daughter cell (n=66, cohorts of 10). Mean value ± SD. (**D**) Graphical schematic of all observed anaphase missegregation events and their frequency. (**E**) Fraction of anaphase missegregation events biased towards the old (red) or new SPB (pink) for mini-chromosome (n=30) and Chr II with normal (Daughter (D)) and inverted segregation of the old SPB (Mother (**M**)) (n=60 and 30). Cohorts of 10–20 missegregating anaphases. Mean value ± SD. (**F**) Frequency of chromosome II loss in *wt*, *Ipl1-321*, at 0 hr (~0–3 completed budding event, CBE), 12 hr (~8–12 CBE), and 24 hr (~18–22 CBE) at 27 °C. Each data point represents loss frequency in a cohort of ~50 cells. Unpaired t-test (*<0.05, **<0.005, ***<0.0005).Mean value ± SD.

The online version of this article includes the following figure supplement(s) for figure 2:

**Figure supplement 1.** Aging sensitizes cells to mild Ipl1 inhibition.

not additive (*Figure 2F*). This suggests that age and the *ipl1-321* mutation affect the same pathway. We concluded that old cells lose chromosomes due to defects in correcting chromosome attachment errors.

## ERC accumulation drives chromosome loss in aging

To identify mechanisms behind the deterioration of mitotic error correction with age, we investigated whether chromosome loss is causally linked to other aging events, particularly to the displacement of the basket from NPCs (*Meinema et al., 2022*). Given that basket displacement is triggered by accumulating ERCs and their SAGA-dependent anchorage to NPCs (*Meinema et al., 2022*; *Denoth-Lippuner et al., 2014*; *Shcheprova et al., 2008*), we first investigated whether ERC accumulation promotes chromosome loss. Mutant cells lacking the deacetylase Sir2, which inhibits recombination in the rDNA locus, form ERCs at an increased rate, accumulate them quicker, and their replicative lifespan is shorter than that of wild-type cells (*Sinclair and Guarente, 1997*; *Kaeberlein et al., 1999*). Supporting the notion that ERC accumulation might promote chromosome loss, these mutant cells also showed a strong and premature loss of the reporter chromosome (*Figure 3A*). To further test the effect of ERC accumulation, we asked whether reducing their formation rate delays chromosome loss. Recombination in the rDNA locus is stimulated by the presence of a replication fork barrier

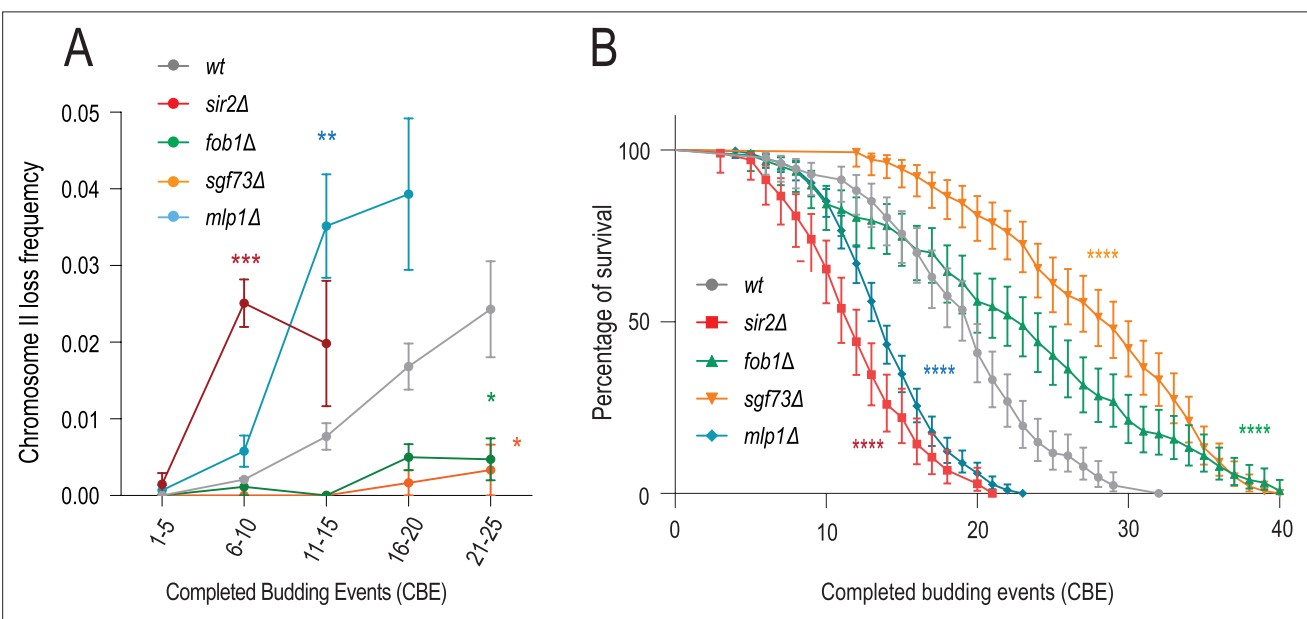

**Figure 3.** extrachromosomal rDNA circle (ERC) formation and nuclear pore complex (NPC) remodeling drive chromosome loss and aging. (**A**) Chromosome II loss frequency per completed budding event (CBE) in strains of indicated genotype (divided into categories of 5xCBE) (*N, n(cells/ divisions)* wt=458/8600, sir2Δ=158/1779, fob1Δ=127/2681, sgf73Δ=142/3950 mlp1Δ=302/4270). Loss frequencies in the same age category are compared to wt. Unpaired t-test (*<0.05, **<0.005, ***<0.0005). Mean value ± SEM (**B**) Replicative lifespan of listed genotypes (n>120 cells, Log-rank (Mantel--Cox) test *<0.05, **<0.005, ***<0.0005).

The online version of this article includes the following figure supplement(s) for figure 3:

**Figure supplement 1.** The effect of introns on chromosome loss is not chromosome-specific.

between rDNA units (*Defossez et al., 1999*). While these barriers prevent the transcription machinery to collide with replication forks coming from neighbouring rDNA units, the stalling forks are fragile, causing high rates of double-strand breaks. Therefore, removing these fork barriers, for example, by inactivating the fork barrier protein Fob1, drastically decreases the rate at which ERCs form and accumulate, stabilizes the nuclear baskets of NPCs, and prolongs the life span of the cells (*Meinema et al., 2022*; *Defossez et al., 1999*). Strikingly, the *fob1Δ* mutant cells trapped in our microfluidics platform lost chromosome II at a substantially reduced frequency compared to wild-type cells, even after dividing 20 times or more (*Figure 3A*). Introducing an artificial replicative DNA circle with a different sequence than an ERC into wild-type cells also promoted chromosome loss, supporting the idea that DNA circles promote chromosome loss with age independently of their sequence (see below). Taken together, the chromosome loss rates seen in cells with artificial DNA circles, along with those in *sir2Δ* and *fob1Δ* cells, show that chromosome loss correlates well with the accumulation of extrachromosomal DNA circles. Since the primary known effect of DNA circle accumulation is to cause the displacement of the nuclear basket from NPCs, we wondered next whether this event was causally linked to chromosome loss. Supporting this notion, removal of the SAGA subunit Sgf73, which mediates ERC attachment to NPCs and the subsequent displacement of their nuclear basket, phenocopied the effect of the *fob1Δ* mutation (*Denoth-Lippuner et al., 2014*): the *sgf73Δ* mutant cells failed to show any significant increase in chromosome loss frequency at any age (*Figure 3A*). Thus, our data suggested that ERC formation and anchorage to NPCs triggered chromosome loss in old yeast cells.

## Nuclear basket remodeling drives chromosome loss

To investigate whether basket displacement promotes chromosome loss in some manner, we next perturbed the nuclear basket and measured whether this accelerated chromosome loss and aging. To do this, we disrupted the *MLP1* gene, encoding the major isoform of the structural basket protein TPR in yeast. In stark contrast to the *sgf73Δ* and *fob1Δ* single mutants and the wild-type cells, the *mlp1Δ* single mutant cells exhibited a rapid increase in chromosome loss frequency as they aged, as determined by using our reporter chromosome II, along with a shortened replicative lifespan (*Figure 3A–B*). Thus, our data suggest that displacement of the basket from NPCs as the cells age promotes chromosome loss through some yet unknown mechanism.

## Introns in genes of the aurora B pathway drive asymmetric chromosome partitioning in aging

Together, our data suggested that the displacement of the nuclear basket from NPCs upon ERC anchorage weakened the molecular pathway that corrected chromosome attachment errors (*Figure 3A*). This impaired ability to correct erroneous chromosome attachments, in turn, caused old mother cells to lose chromosomes to their buds. To test this hypothesis, we next investigated how basket displacement could lead to asymmetric chromosome partitioning. The basket of the nuclear pore has been involved in various processes, such as mRNA export (*De Magistris, 2021*), protein quality control (*Niepel et al., 2013*), docking of environmentally regulated genes to the nuclear periphery upon their induction (*Raices and D'Angelo, 2017*), the retention of faulty mRNAs in the nucleus, such as unspliced pre-mRNAs (*Galy et al., 2004*; *Bonnet and Palancade, 2015*), as well as docking Mad1 and Mad2, two key components of the mitotic spindle assembly checkpoint (SAC), to the NPC (*Iouk et al., 2002*; *Scott et al., 2005*).

The SAC components Mad1 and Mad2 bind Mlp1, Mlp2, and Nup60 at the basket of NPCs (*Scott et al., 2005*). However, basket dissociation, which delocalizes Mad1/2 from the nuclear periphery, does not appear to interfere with their function (*Scott et al., 2005*). Nevertheless, we investigated whether SAC dysfunction could account for aging phenotypes caused by basket dissociation and for the premature aging phenotype seen in *mlp1Δ* mutant cells. Arguing against this hypothesis, the *mad1Δ* and *mad2Δ* single mutant cells exhibited no change in replicative lifespan compared to wild-type cells and did not phenocopy *mlp1Δ* mutant cells (*Figure 3—figure supplement 1A*). Therefore, we searched for other mechanisms through which basket displacement could trigger chromosome loss.

While perusing the list of genes involved in chromosome segregation (*Figure 4A*), we noticed that only three of them contain an intron, *NBL1*, *MCM21*, and *GLC7*. Surprisingly, these three genes all function with Ipl1/aurora B in preventing syntelic chromosome attachment. Nbl1/borealin is a

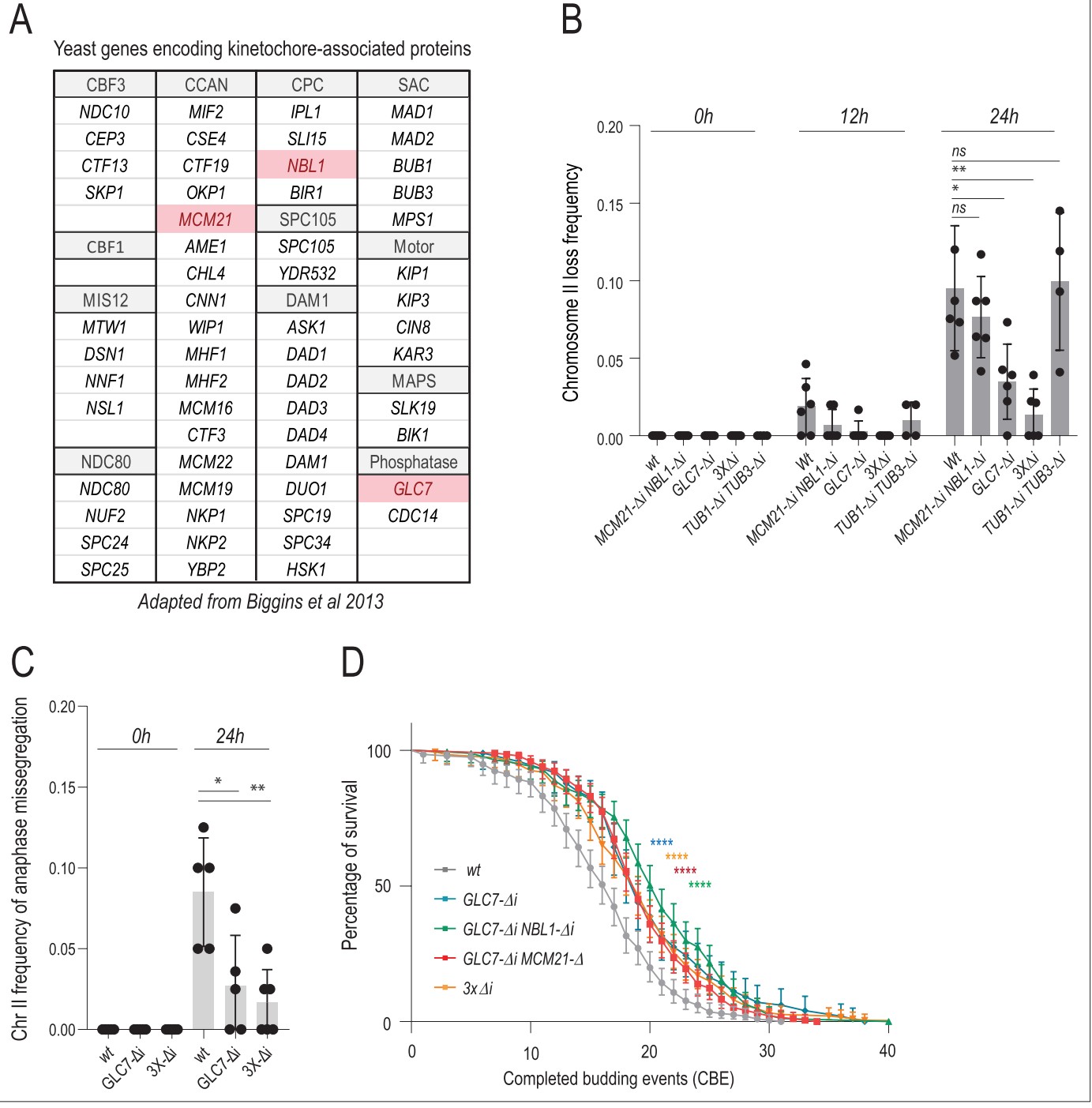

**Figure 4.** Introns drive asymmetric chromatid partitioning and chromosome loss in aging. (**A**) A list of genes encoding kinetochore-associated proteins adapted from *Biggins, 2013* . Intron-containing genes are marked in red. (**B**) Chromosome II loss frequency at indicated aging time (top) in cells of indicated genotype. *3xΔi stands for GLC7-Δi MCM21-Δi NBL1-Δi* triple mutant. Each data point represents the frequency of chromosome loss in a cohort of ~50 cells. Unpaired t-test (*<0.05, **<0.005, ***<0.0005). Mean value ± SD. (**C**) Frequency of anaphase missegregation of chromosome II at indicated aging time (top) in cells of indicated genotype. *3xΔi stands for GLC7-Δi MCM21-Δi NBL1-Δi* triple mutant. Each data point represents missegregation frequency in a cohort of ~50 anaphases. Loss frequencies in the same age category are compared to wt. Unpaired t-test (*<0.05, **<0.005, ***<0.0005). Mean value ± SD. (**D**) Replicative lifespan upon intron removal (n>200 cells, Log-rank (Mantel--Cox) test, *<0.05, **<0.005, ***<0.0005).

The online version of this article includes the following source data and figure supplement(s) for figure 4:

**Figure supplement 1.** The 3x *Δi*removal does not alter mitotic timing, growth or budding in old age due to *GLC7Δi* duplication.

**Figure supplement 1—source data 1.** Uncropped gel image used to generate *Figure 4—figure supplement 1F*.

*Figure 4 continued on next page*

*Figure 4 continued*

**Figure supplement 1—source data 2.** Uncropped gel image used to generate *Figure 4—figure supplement 1F* with labels and cropped area marked.

component of the chromosome passenger complex of which aurora B/Ipl1 is the catalytic subunit. Mcm21/CENP-O is a docking receptor for Ipl1/aurora B at kinetochores. Glc7/PP1A is the protein phosphatase 1, which counteracts aurora B by dephosphorylating its targets at the outer-kinetochore (*Pinsky et al., 2006*; *Poddar et al., 1999*; *Nakajima et al., 2009*). Intrigued by this remarkable convergence, we wondered whether asymmetric chromosome partitioning in aging could stem from the presence of these three introns in error correction genes. We rationalized that displacement of the basket from NPCs could potentially affect the regulation, export, and hence expression of intron-containing transcripts, specifically, thereby affecting the function of the *NBL1*, *MCM21*, and *GLC7* genes.

To investigate the possibility that they contribute to the high rate of chromosome loss in old cells, we removed the introns of *GLC7*, *MCM21*, *NBL1* (*-Δi* alleles), and imaged these mutant cells throughout their replicative lifespan, monitoring chromosome II segregation. Removing each intron individually or all three simultaneously (*3 xΔi* strain) had no detectable effects on chromosome segregation in young cells (*Figure 4B*), cell growth and mitotic timing (*Figure 4—figure supplement 1A–C*). Interestingly, whereas the removal of *GLC7*'s intron decreases the expression of the gene by about 50% and affects cell proliferation (*Parenteau et al., 2019*; *Parenteau et al., 2008*; *Juneau et al., 2006*), we note that the *GLC7-Δi* allele is duplicated in our strain, probably owing to recombination between transposon LTRs surrounding it (*Figure 4—figure supplement 1E–F*). We suspect that this duplication explains why our *GLC7-Δi* mutant strain grew as well as wild-type cells. Simultaneously removing all three introns nearly completely suppressed chromosome loss in old cells (*Figure 4B–C*). Removing them individually or in pairs did so as well, albeit the effects were milder. Importantly, the simultaneous removal of *NBL1*, *MCM21*, and *GLC7* introns did not affect the duration of the last budding event (*Figure 4—figure supplement 1D*), indicating that it suppressed chromosome loss without altering cell cycle progression. Removing the introns of the two α-tubulin genes *TUB1* and *TUB3* simultaneously had no effect, indicating that the observed effects were specifically related to the introns of *NBL1*, *MCM21*, and *GLC7* and not to introns in general (*Figure 4B*). The effects observed were similar for chromosome II and chromosome IV, indicating that the role of these introns was not chromosome specific (*Figure 3—figure supplement 1B*). Furthermore, removing these three introns also suppressed the occurrence of asymmetric anaphases in old cells (*Figure 4C*), supporting the view that asymmetric chromosome segregation and chromosome loss go hand-in-hand and are under the control of these three introns. Strengthening this conclusion, removal of these introns also suppressed chromosome loss in the *ipl1-321* mutant cells (*Figure 2—figure supplement 1B*).

Remarkably, simultaneous removal of *MCM21*, *NBL1*, and *GLC7* introns also had a substantial effect on the longevity of the cells, extending their replicative lifespan up to ~20% (*Figure 4D*). A similar effect was observed in the *ipl1-321* mutant cells (*Figure 2—figure supplement 1A*). Thus, chromosome loss may indeed be the first direct cause of death for many yeast cells upon aging, although other mechanisms eventually take over when chromosome loss is abrogated.

## Basket destabilization causes intron-dependent chromosome loss in young cells

Our data were consistent with the hypothesis that basket dissociation from NPCs promotes chromosome loss through an intron-dependent dampening of the error correction pathway. To test this hypothesis in greater depth, we asked whether mutations affecting the nuclear basket also promote chromosome loss outside of the aging context. Thus, we characterized chromosome segregation in young *mlp1Δ* cells (*Figure 5A–C*). Analysis of chromosome II segregation indicated that the *mlp1Δ* mutant cells not only exhibited a fourfold increase in their chromosome missegregation frequency compared to the young wild-type cells, but ~80% of these missegregation events followed the old SPB, as in old cells (*Figure 5C*). Furthermore, this chromosome loss was suppressed upon removal of the introns of *MCM21*, *NBL1*, and *GLC7* (*Figure 5B*). Therefore, although to a lower extent than in old cells, defects in the nuclear basket result in asymmetric chromosome partitioning in an intron-dependent manner in young cells as well. Interestingly, removal

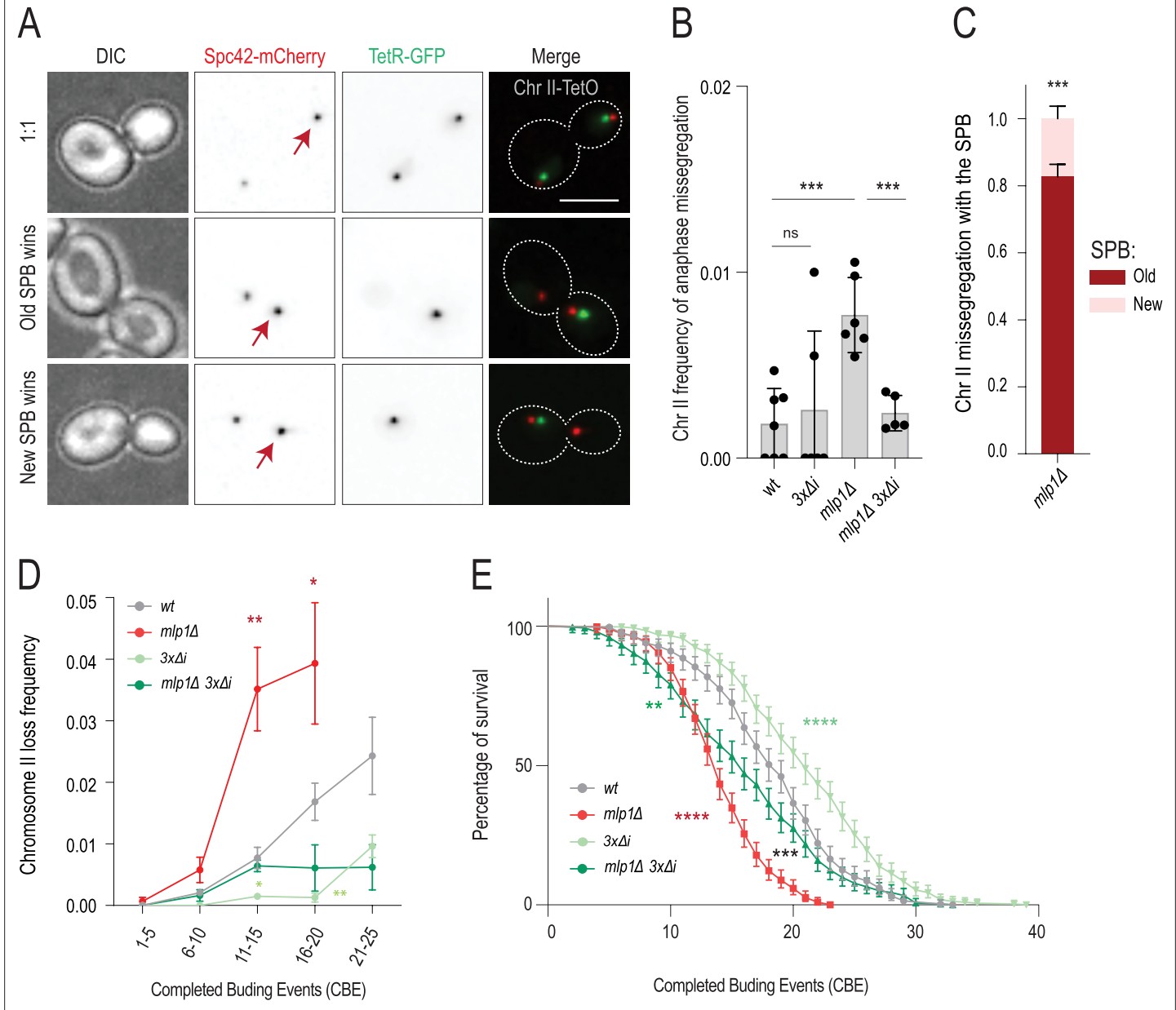

**Figure 5.** Basket displacement causes intron-dependent chromosome loss in young and old cells. (**A**) Stills of symmetric and asymmetric anaphases in young cells. Arrow marks the old spindle pole body (SPB). Scale bar (upper right panel) is 5 µm. (**B**) Frequency of anaphase missegregation of chromosome II in cells of indicated genotypes (*n>2500 cells*). Each data point represents anaphase missegregation frequency in a cohort of >300 cells. Mean value ± SD. (**C**) Fraction of chromosome II missegregation with old (red) or new SPB (pink) in anaphase cells of indicated genotype (n=30 cohorts of ~10 missegregation events) Unpaired t-test (*<0.05, **<0.005, ***<0.0005). Mean value ± SD. (**D**) Chromosome II loss frequency in indicated genotype as a function of CBE (categories of 5xCBE) (*N, n(cells/divisions) wt=458/8600, mlp1Δ=302/4270,3 xΔi=469/9974 mlp1Δ 3 xΔi=295/4757*).Mean value ± SEM. (**E**) Replicative lifespan of listed genotypes. Red and green stars represent *p-values* between the wt and the corresponding genotype. Black stars represent the p-value between *mlp1Δ* and *mlp1Δ 3 xΔi* (n>300 cells, Log-rank (Mantel--Cox) test, *<0.05, **<0.005, ***<0.0005).

of these three introns also rescued the age-dependent chromosome loss and partially restored the longevity of the *mlp1Δ* mutant cells (*Figure 5D–E*). We concluded that intron-dependent dampening of the error correction pathway is one, even if not the only progeric effects of the *mlp1Δ* mutation. Possibly, other introns promote additional aging phenotypes. Thus, we conclude that basket displacement promotes chromosome loss in young mutant cells and old wild-type mother cells.

## Intron deletion suppresses the effect of DNA circle accumulation on chromosome loss

As seen above, chromosome loss correlates well with conditions that promote ERC accumulation and their anchorage to NPCs. This idea gained in traction as we progressed, because ERC attachment, and the attachment of any DNA circle, to NPCs is known to mediate nuclear basket displacement (*Meinema et al., 2022*). Thus, we further investigated the relationships between DNA circle accumulation and chromosome loss by investigating how the removal of *NBL1*, *MCM21,* and *GLC7* introns affected chromosome loss observed upon DNA circle accumulation. For this, we followed two parallel approaches. First, we tested whether introducing a replicative DNA circle into the cells is sufficient to drive chromosome loss during aging, and if so, whether this event required the presence of the introns of *NBL1*, *MCM21*, and *GLC7*. Thus, we transformed the wild-type and the *3 xΔi* mutant strains carrying the labeled chromosome II as reporter with the yeast replicative plasmid YRp17 (*Mann and Davis, 1983*). We then monitored the frequency at which these transformed cells lost the reporter chromosome as they underwent aging. YRp17 contains a replication origin (*ARS1*) but no centromere, such that it accumulates in the mother cells with each replicative cycle (*Figure 6A*). As expected (*Sinclair and Guarente, 1997*), the cells carrying YRp17 showed a much shorter lifespan (10 CBE median RLS, *Figure 6B*). Remarkably, however, the effect of YRp17 on longevity was substantially suppressed by the 3 xΔi mutations (*Figure 6B*). Analysis of chromosome loss indicated that introducing this DNA circle promoted chromosome loss, starting early in life and increasing rapidly with age (*Figure 6C*). Furthermore, removal of the introns of *NBL1*, *MCM21*, and *GLC7* (*3 xΔi* cells) largely suppressed the effect of YRp17 on chromosome loss. However, note that the suppressive effect of the *3 xΔi* mutations on longevity and chromosome loss was only partial, indicating that YRp17 promotes chromosome loss through both a mechanism that involves these three introns, and at least one additional mechanism independent of these introns.

As a second approach, we investigated whether removing the introns of *NBL1*, *MCM21,* and *GLC7* mitigates the chromosome loss phenotypes of the *sir2Δ* mutant cells. Supporting this hypothesis, the *3 xΔi* triple mutation significantly, though modestly, improved the longevity phenotype of these cells (*Figure 6D*). As in cells carrying YRp17, the triple intron removal indeed ameliorated the premature chromosome loss phenotype of the *sir2Δ* mutant cells (*Figure 6E*). As for YRp17, the effect of the 3 xΔi was substantial and significant, but partial. Thus, we concluded that the accumulation of DNA circles, such as ERCs and others, which promote the dissociation of the basket from NPCs and thereby drive chromosome loss at least in part through the introns of *NBL1*, *MCM21,* and *GLC7*. While this effect appeared to be the dominant one in physiologically aged cells, additional effects seem to further promote chromosome loss in the *sir2Δ* mutant cells and cells carrying an exogenous DNA circle, such as YRp17.

## Old cells leak pre-mRNA to the cytoplasm

Putting our different observations together, our data suggested that the removal of the nuclear basket from NPCs in response to the accumulation of DNA circles with age causes old yeast mother cells to lose chromosomes to their last bud. Furthermore, the effect of nuclear basket removal from NPCs was primarily mediated by three introns. This suggested that the nuclear basket of NPCs enforces proper chromosome segregation through its function in pre-mRNA processing or quality control. To test this possibility, we next investigated whether old mother cells show defects in pre-mRNA quality control. Since basket mutant cells fail to retain unspliced pre-mRNAs in the nucleus (*Galy et al., 2004*; *Bonnet and Palancade, 2015*), we reasoned that old yeast mother cells also might leak pre-mRNAs to the cytoplasm.

As a first proof of principle, we characterized the localization of intronic RNA sequences in young and old cells, using single-molecule RNA fluorescence in situ hybridization (smRNA FISH). Most yeast introns are short (50–150 nucleotides) and well below the minimal size detectable by smRNA FISH (>500 nts, ideally >1 kb). In the yeast genome, only 20 introns are longer than 500 nucleotides. Thus, we created probes for the longest introns, namely those of *DBP2* (766 nts) and *YRA1* (1002 nts), and of *GLC7* because of its role in chromosome segregation (525 nts). The introns of *NBL1* (66 nts) and *MCM21* (88 nts) are too short for smRNA FISH. Cohorts of old cells were collected using the mother enrichment program (MEP) *Lindstrom and Gottschling, 2009* followed by FACS sorting to reach high homogeneity (see methods). By inducing cell death in the daughter cells specifically, the MEP ensures

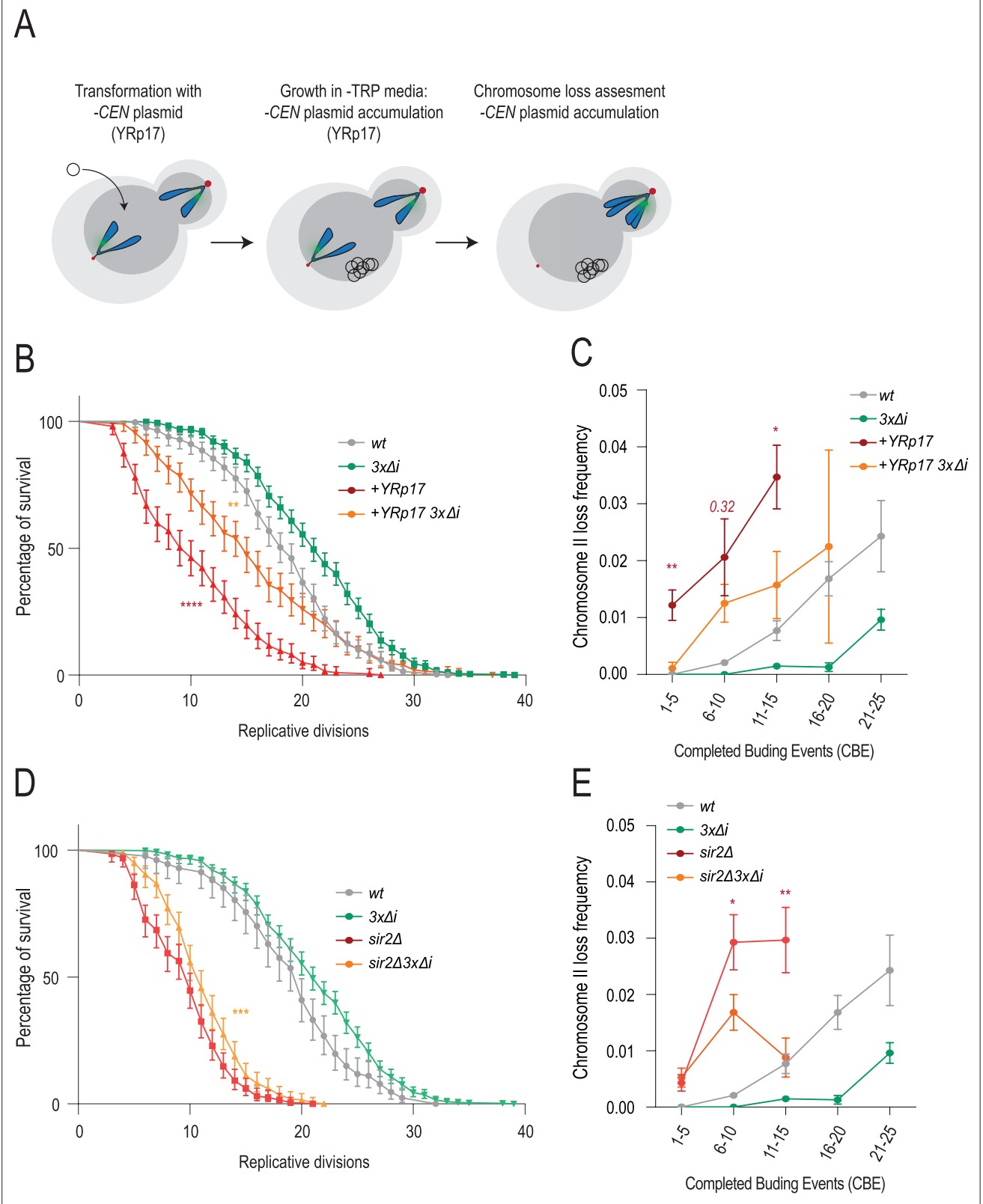

**Figure 6.** Non-centromeric DNA is sufficient to drive aging and chromosome loss. (**A**) Schematic of transformation of chromosome II reporter strains with YRp17 (-*CEN* plasmid) and the accumulation of non-centromeric plasmids. (**B**) Replicative lifespan of strains with and without YRp17 plasmid. Red stars represent *p-values* between the YRp17 and YRp17 *3 xΔi* strain. Orange stars represent the *p-value* between YRp17 *3xΔi* and wt (n>200 cells, Log-rank (Mantel--Cox) test, *<0.05, **<0.005, ***<0.0005). (**C**) Chromosome II loss frequency in indicated genotypes as a function of completed budding

*Figure 6 continued on next page*

*Figure 6 continued*

event (CBE) (divided into categories of 5xCBE) (*N, n(cells/divisions) wt=458/8600, 3 x∆i=469/9974, +YRp17=199,2097, +YRp17 3 x∆i=200,3134*). Mean value ± SEM. (**D**) Replicative lifespan of *sir2∆ and sir2∆3x∆i* strain. Stars represent *p-value* between *sir2∆ and sir2∆3x∆i* (n>200 cells, Log-rank (Mantel-Cox) test, \*<0.05, \*\*<0.005, \*\*\*<0.0005). (**E**) Chromosome II loss frequency in indicated genotype as a function of CBE (divided into categories of 5xCBE) (*N, n(cells/divisions) wt=458/8600, 3 x∆i=469/9974, sir2∆=351,3500, sir2∆3x∆i=350,3840*). Mean value ± SEM.

that aging mother cells are diluted linearly instead of exponentially in the population, greatly facilitating the isolation of very old cells. FACS sorting ensured that the cohorts were free of young cells and young cell debris. With each of these three series of smRNA FISH probes, we detected one or two nuclear foci in young cells (*Figure 7A*). Only very few cells (<3%) showed a focus in the cytoplasm. No foci were detected upon deleting the corresponding introns (∆i), demonstrating probe specificity (*Figure 7—figure supplement 1A–B*). In contrast, for each of the three introns tested, smRNA FISH detected intronic RNA sequences in the cytoplasm of most old cells. This was not due to an increase in overall smRNA FISH foci number during aging (See *Figure 7—figure supplement 1C*), but due to the change in foci distribution between nucleus and the cytoplasm. For example, in the case of *GLC7*, ~70% of smRNA FISH foci were found in the cytoplasm of old mother cells, compared to ~3% in young cells (*Figure 7A–B*). Thus, 3 out of 3 tested introns showed a strong tendency to leak to the cytoplasm in old mother cells. However, this first test did not clarify whether the full pre-mRNA or only the intron escaped from the nucleus.

To directly address this question, we turned to a functional assay and monitored the translation of an unspliced pre-mRNA reporter in the cytoplasm. This fluorescent reporter contains the coding sequence for mCherry in the intron and expresses GFP upon proper splicing of the nascent transcript. Therefore, mCherry is only translated when the unspliced pre-mRNA leaks out of the nucleus (*Figure 7C*; *Sorenson and Stevens, 2014*). Cells carrying this construct integrated at the *TRP1* locus were trapped and imaged in our microfluidics platform. As cells aged, we measured the number of completed budding events and the fluorescence levels of the two reporter fluorophores. Fluorescence signal analysis established that the mCherry/GFP ratio increased in old compared to young cells, demonstrating elevated pre-mRNA translation in the cytoplasm of a large majority of aged cells (*Figure 7D–E*).

Consistent with ERC-dependent leakage and translation of this reporter, the mCherry/GFP ratio increased prematurely and to a higher level in the *sir2∆* mutant, compared to the wild-type cells (*Figure 7D–F*). The fact that the signal was already observable in young *sir2∆* mutant cells suggests that the pleiotropic functions of Sir2 cause a fraction of these effects. Conversely, the *fob1∆* mutant cells showed a lowered mCherry/GFP ratio, which did not increase much during their lifespan (*Figure 7D–F*). Thus, we conclude that ERC formation and accumulation promote pre-mRNA leakage during yeast aging, as expected from their effect on the nuclear basket.

## Pre-mRNA leakage causes intron-dependent chromosome loss in young cells

Given the results above, we wondered whether chromosome loss in old cells is linked to their failure to retain pre-mRNAs in the nucleus. Thus, we asked whether inducing pre-mRNA leakage in young cells would trigger chromosome loss. The two yeast SR-proteins Gbp2 and Hrb1 have been implicated in the retention of pre-mRNAs in the nucleus by binding intron-containing transcripts and delaying their export until they are properly spliced or degraded (*Hackmann et al., 2014*; *Hurt et al., 2004*). Accordingly, the *hrb1∆ gbp2∆* double mutant cells leak pre-mRNA to the cytoplasm at a higher rate than wild-type cells (*Hackmann et al., 2014*; *Windgassen et al., 2004*). Thus, we asked whether these double mutant cells lost the labeled chromosome II at an increased frequency. As a control, we asked whether the *snu66∆* mutation, which affects the spliceosome itself directly (*Stevens and Abelson, 1999*; *Murphy et al., 2004*), lead to similar phenotypes.

The young spliceosome-defective *snu66∆* mutant cells did exhibit a fivefold elevated level of chromosome missegregation compared to young wild-type cells. However, this missegregation was not biased towards any specific SPB (*Figure 8A–B*), suggesting that it involved a distinct mechanism than that observed in old cells and in young cells lacking the basket protein Mlp1. In contrast to the *snu66∆* mutant, the *hrb1∆ gbp2∆* double mutant cells showed an even further increased frequency of chromosome missegregation, which was more than 10-fold above that of young wild-type cells

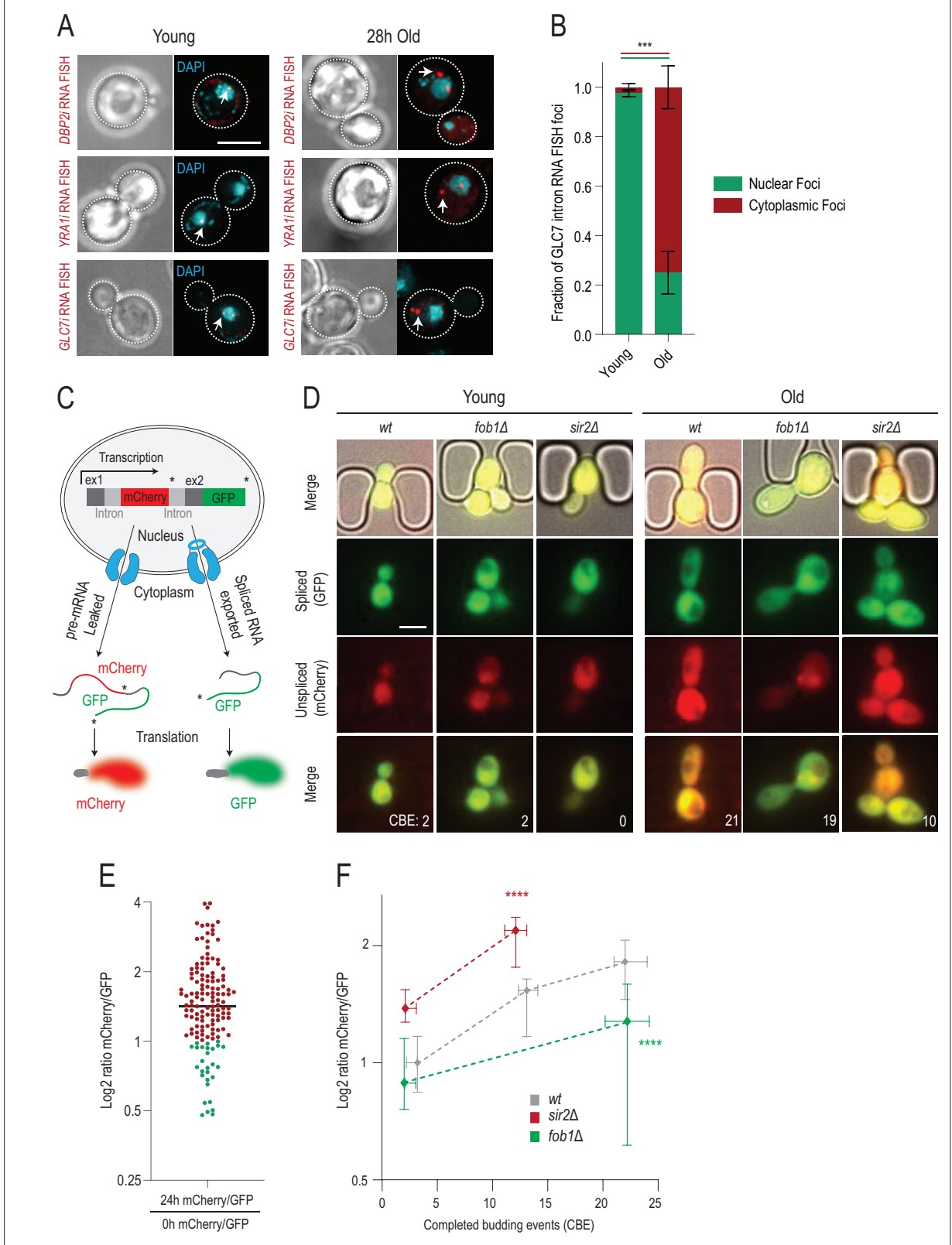

**Figure 7.** Pre-mRNA leaks to the cytoplasm of old cells. (**A**) Images of RNA fluorescence in situ hybridization (RNA FISH) targeting three long introns (*GLC7, YRA1, DBP2*) in young and old cells. Scale bar (upper left panel) is 5 μm. (**B**) Quantification of *GLC7* intron RNA FISH foci localization in young and old cells (n=295, cohorts of ~100 cells). Mean value ± SD. (**C**) Principle of the pre-mRNA translation reporter (left panel) adapted from *Sorenson and Stevens, 2014*. (**D**) Images of the pre-mRNA translation reporter in cells (right panel) of indicated genotype (top) at indicated ages (bottom). Scale bar

*Figure 7 continued on next page*

*Figure 7 continued*

(upper left panel) is 5 µm. (**E**) Ratio of mCherry/GFP signal in the same cells at 0 hr and after 24 hr of aging. 119/146 cells have a ratio greater than '1.' (**F**) Log2 ratio of mCherry/GFP in cells of indicated genotype as a function of age, normalized to wt median at 0 hr (n=60, 35, 32). Unpaired t-test comparing mCherry/GFP intensity of cells at 12 or 24 hr (*<0.05, **<0.005, ***<0.0005).Mean value ± SD.

The online version of this article includes the following figure supplement(s) for figure 7:

**Figure supplement 1.** Single molecule RNA fluorescence in situ hybridization (smRNA FISH) probes are intron-specific.

(**Figure 8A**). Furthermore, sister chromatid missegregation was strongly biased towards the old SPB (**Figure 8B**). Removing the introns of *GLC7*, *MCM21*, and *NBL1* suppressed this chromosome segregation defect (**Figure 8A**). Therefore, pre-mRNA leakage is sufficient to induce asymmetric chromosome partitioning and is contingent on the presence of introns in the error correction genes *NBL1*, *MCM21*, and *GLC7*. The fact that mutations affecting the spliceosome did not promote asymmetric chromosome segregation to any specific SPB with age indicates that the chromosome loss observed in old cells is not due to splicing defects.

We conclude that the accumulation of extrachromosomal DNA causes the displacement of the NPC basket and subsequent leakage of pre-mRNAs. Leakage of intron-containing transcripts into the cytoplasm impairs correction of syntelic attachments by the Ipl1/aurora B pathway, likely because three genes involved in this pathway contain introns. This results in asymmetric partitioning of sister chromatids and age-associated chromosome loss. In summary, basket displacement and pre-mRNA leakage impair error correction via an as-yet unknown mechanism involving the introns of *NBL1*, *MCM21* and *GLC7*.

## Discussion

Aging manifests itself through a broad diversity of conserved cellular phenotypes, but in most cases, we know little about how these are causally linked to each other (**Denoth Lippuner et al., 2014**;

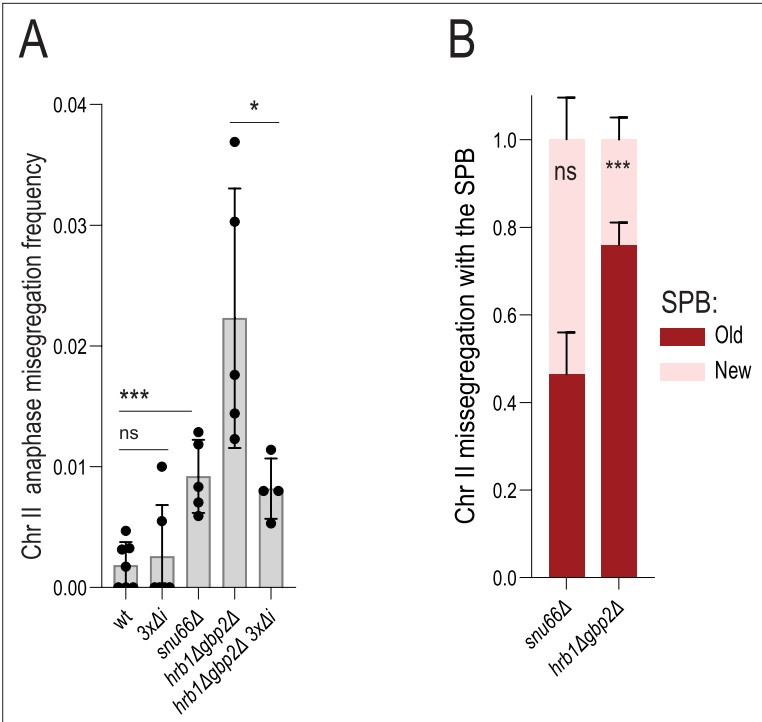

**Figure 8.** Pre-mRNA leakage is sufficient to drive asymmetric chromatid partitioning. (**A**) Chromosome II missegregation in cells of indicated genotypes (*n=2774, 3000, 5269, 2650, 1607*). Each data point represents anaphase missegregation frequency in the cohort of >300 cells. Mean value ± SD. (**B**) Fraction of chromosome II missegregation with old (red) or new spindle pole body (SPB) (pink) in anaphase cells of indicated genotype (n=50,30 cohorts of ~10 missegregation events). Unpaired t-test (*<0.05, **<0.005, ***<0.0005).Mean value ± SD.

*López-Otín et al., 2023*; *Janssens and Veenhoff, 2016*). In this study, we investigated whether NPC remodelling in old yeast cells contributed to the emergence of other aging phenotypes, using chromosome loss as a case study (*Meinema et al., 2022*). We showed that NPC remodelling is both necessary and sufficient to drive the age-associated increase in chromosome loss and we identified the mechanisms involved. Specifically, interventions that promoted displacement of the NPC basket, or basket defects more broadly, enhanced non-disjunction and missegregation of sister chromatids to the bud, thereby promoting chromosome loss in old yeast cells. These interventions included mutations that stimulated ERC formation and removed the main yeast TPR isoform, Mlp1. Conversely, interventions that prevented NPC remodelling, such as delaying ERC accumulation or preventing their anchorage and the recruitment of SAGA to NPCs, restored proper chromosome segregation in old cells and suppressed chromosome loss. Thus, NPC remodeling appeared to be the pivotal point for triggering chromosome loss in old cells.

Two potential caveats should be considered. First, the *sir2Δ* and *fob1Δ* mutations each have effects beyond influencing ERC formation rates. The *sir2Δ* mutant cells lose silencing of the hidden mating type loci, genetically behaving as diploids. They fail to silence subtelomeric regions and act through some as-yet unknown mechanism on the maintenance of proteostasis (*Haber, 2012*; *Rine and Herskowitz, 1987*; *Gottschling et al., 1990*; *Smith and Boeke, 1997*). Likewise, the *fob1Δ* mutation renders the rDNA locus unstable, which might interfere with rRNA transcription and ribosome biosynthesis (*Kobayashi et al., 1998*). However, their only clear common denominator is that they both affect ERC formation, and hence NPC remodelling, even if in opposite manners (*Kaeberlein et al., 1999*; *Defossez et al., 1999*). Moreover, the *sgf73Δ* mutation phenocopies the effects of the *fob1Δ* mutation. Here again, the only known functional overlap between Fob1 and Sgf73 is that they promote ERC accumulation in the mother cell indirectly (Fob1) or directly (Sgf73), ERC anchorage to NPCs, and the consequent displacement of NPC basket (*Meinema et al., 2022*; *Denoth-Lippuner et al., 2014*). Finally, introducing another DNA circle in the form of a non-centromeric replicative plasmid, a process that also causes basket displacement (*Meinema et al., 2022*), promoted chromosome loss as well. Thus, our data establish that DNA circle accumulation and attachment to NPCs trigger chromosome loss. Destabilizing the basket independently of DNA circle accumulation also promotes the premature onset of chromosome loss, indicating that it is the basket displacement from NPCs that triggers chromosome loss upon DNA circle accumulation.

A second possible caveat is that chromosome loss could be linked to aging in some other manner. To address this possibility, we investigated the molecular mechanisms that link basket displacement and chromosome loss. Our data reveal that chromosome loss is due to the non-correction of syntelic chromosome attachments in old cells. In young cells, the kinase aurora B and its accessory factors enforce such corrections (*Biggins et al., 1999*). Our data also show that the inhibition of error correction in old cells requires the presence of three introns, those of the genes *NBL1*, *MCM21*, and *GLC7*. These three genes are the only chromosome segregation machinery genes that contain an intron. Strikingly, the proteins Nbl1, Mcm21, and Glc7 are all involved in the error correction pathway, together with aurora B/Ipl1 (*Pinsky et al., 2006*; *Poddar et al., 1999*; *Nakajima et al., 2009*). Removing their introns restored proper attachment correction and chromosome segregation in old cells and suppressed the premature chromosome loss phenotype of the *ipl1-321* mutant cells grown at semi-permissive temperature. Intron removal delays cell death but does not abrogate aging. Furthermore, removing these introns not only suppresses chromosome missegregation in old wild-type cells, but also in young *mlp1Δ* mutant cells. Thus, intron removal separates aging and chromosome loss, indicating that these introns are part of the proximal cause of chromosome loss and act independently of age. Therefore, our data indicate that there is a direct, intron-dependent link between basket displacement from NPCs and chromosome loss. Our data indicate that this link involves the leakage of unspliced pre-mRNAs out of the nucleus upon basket removal and affects the basket's function in mRNA quality control. The fact that another independent set of mutations that also interfere with pre-mRNA retention in the nucleus, such as the *gbp2Δ hrb1Δ* double mutant cells, also increases the frequency of chromosome loss through the same three introns strengthens this conclusion. Thus, together our data establish that displacement of the nuclear basket from NPCs in old mother cells directly triggers chromosome non-disjunction and loss through causing the leakage of the pre-mRNAs of the intron-containing genes *NBL1*, *MCM21*, and *GLC7*.

This conclusion has several consequences and opens new questions. The key question stems from our observation that releasing pre-mRNA from the nucleus relaxes the control that cells impose on chromosome attachment. The question is, how does this work? Does it involve the translation of the pre-mRNAs in the cytoplasm? Does it come about through the correlated loss of properly mature mRNAs and hence, the reduced expression of their products? Two arguments speak against these interpretations. First, if any of these two options were to account for the effect of pre-mRNA leakage, mutations affecting splicing efficiency would lead to the same effects. However, tempering with the spliceosome does not affect the correction of chromosome attachment errors to any extent and specificity close to the effects of pre-mRNA leakage. Second, since Mcm21 and Glc7 have opposite effects on chromosome correction, it is unclear how impairing the expression of both would lead to the synergistic effects that we observe between them. Finally, analysis of RNA-seq data from replicatively aged cells indicates that if aging indeed impairs the splicing of some transcripts, such as those encoding ribosomal proteins, it stimulates that of others (*Gómez-Montalvo et al., 2024*). This suggests that aging has a non-linear effect on intron-containing transcripts. It will be important to determine whether basket displacement contributes to these effects, how it does so, and whether this could explain the effect it has on the error correction pathway.

Our observation that basket displacement has such profound and highly specific impacts on processes functionally very distant from that of the NPC, such as chromosome segregation, has several consequences as well. First, it means that NPC remodelling could affect many more cellular functions than only chromosome segregation. Interestingly, beyond mRNA quality control, the basket of the NPC is also involved in protein quality control and the activation of environmentally regulated genes, two processes that are impaired in old cells (*Denoth Lippuner et al., 2014*; *López-Otín et al., 2023*). Thus, basket displacement might contribute to these phenotypes in aging cells. It is also very likely that the role of the basket in mRNA quality control could affect more than only chromosome segregation. Curiously, while budding yeasts have kept only a subset of the introns of their ancestors, these introns are not distributed randomly and seem to be enriched in very specific processes, such as genes coding for ubiquitin-conjugating enzymes and proteasome subunits, components of the mitochondrial respiratory chain, and ribosomal proteins (*Lim et al., 2021*). Interestingly, decay of the ubiquitin-mediated protein degradation, mitochondrial membrane potential, and ribosome assembly are classical and ubiquitous hallmarks of aging (*Denoth Lippuner et al., 2014*; *López-Otín et al., 2023*; *Janssens and Veenhoff, 2016*). Thus, it is possible that basket displacement in old cells contributes to the emergence of these aging phenotypes as well.

A second type of consequences emerging from our observations stems from the fact that displacement of the NPCs' basket is also observed in stressed cells and not solely in the context of aging. For example, heat shock also causes basket displacement in yeast (*Carmody et al., 2010*). Therefore, it is possible that the downstream effects we observe in old cells also take place during stress response. In this respect, it is interesting to note that heat shock has been associated with a strong rise in aneuploidy in yeast (*Yona et al., 2012*; *Chen et al., 2012*; *Shen et al., 2020*). Furthermore, aneuploidy is a common response to environmental stress and a primary mechanism for the emergence of stress-resistant clones in fungi (*Gilchrist and Stelkens, 2019*). Therefore, it is tempting to speculate that the chromosome missegregation resulting from basket displacement might be a selected response to stress. It might be selectively advantageous for cells to actively promote karyotype variations when their survival is in question. It is therefore worth considering that a similar process, perhaps using similar mechanisms, underlies the aneuploidy of cancer cells.

Third, the role of basket displacement in the emergence of aging phenotypes is likely not a peculiarity of yeast. Using proteomics, basket displacement has been inferred to take place in aging cells such as the hepatocytes of old mice and humans (*Ori et al., 2015*; *Malik et al., 2023*). Little is known about whether the nuclear basket of metazoans plays a similar role in pre-mRNA retention in the nucleus as that of yeast, but it is striking that intron retention is one of the hallmarks of aging on the metazoan transcriptome, and that pre-mRNA processing genes are important for mitotic stability (*Mariotti et al., 2022*; *Adusumalli et al., 2019*; *Huang et al., 2022*; *Funk et al., 2022*). Intron retention would be expected if some transcripts escaped the nucleus prematurely. More strikingly, perhaps the most ubiquitous effect of the expression of the progeric variant of lamin A, progerin, across cell types is the displacement of the nuclear basket protein TPR from NPCs (*Larrieu et al., 2018*). We know still very little about the impact of basket displacement in the aetiology of the Gilford-Hutchinson

Progeria Syndrome, but our study suggests that it might be relevant. If so, investigating the potential effects of progerin on pre-mRNA leakage might become an essential step towards understanding how progerin expression causes premature aging.

In a broader sense, our findings might bring fundamental insights towards our understanding of aging. For decades, the role of ERCs in aging has been perceived as a yeast peculiarity. However, the effect it has on NPCs may close the gap. On one hand, it seems to illuminate what might be the commonality between ERC-driven aging in yeast and other aging processes in other eukaryotes. Indeed, the basket-centric view of aging that we propose here would explain how ERCs actually cause aging in yeast and indicate that while ERC accumulation might be a yeast idiosyncrasy, its downstream effects are not. On the other hand, this idea opens two corollary questions. (1) What is the biological significance of eccDNAs promoting basket displacement in yeast? (2) What causes basket displacement in other aging systems? We are intrigued by the idea that ERCs and any replicating eccDNA for that matter are the closest mimics that laboratory yeast strain may encounter of pathogenic DNA of exogenous origin. In that perspective, basket displacement and the ensuing effects might have first appeared as a response to infections. In other words, yeast aging might reflect the activity of a death pathway that normally functions in pathogen control. It might represent a fungal version of the inflammatory or auto-immune response in metazoans. Therefore, it will be interesting to investigate whether exogenous or non-chromosomal DNA, such as that of viruses, also triggers the displacement of the nuclear basket from NPCs in yeast and other systems, and whether this contributes to either or both of anti-viral defence and aging. If so, aging might have emerged as an anti-viral defence.

# Materials and methods

**Key resources table**

| Reagent type (species) or resource | Designation | Source or reference | Identifiers | Additional information |
|---|---|---|---|---|
| Genetic reagent (*Saccharomyces cerevisiae*) | YYB strains | This paper | YYB | Full list in *Supplementary file 1* |
| Recombinant DNA reagent | CEN TetO-plasmid | *Denoth-Lippuner et al., 2014* | YYB 16094 strain | |
| Recombinant DNA reagent | CEN Ubi-GFP plasmid | This paper | PYB2665 | Sequence in *Figure 1—source data 1* |
| Recombinant DNA reagent | YRp17 | *Mann and Davis, 1983* | PYB546 | Sequence in *Figure 1—source data 1* |
| Sequence-based reagent | MM Primers | This paper | MM primers | Full list in *Supplementary file 1* |
| Sequence-based reagent | smRNA FISH probes | This paper | *GLC7, DBP2, YRA1* smRNA FISH probes | Full list in *Supplementary file 1* |
| Commercial assay, kit | QIAGEN gel extraction kit | QIAGEN | Ref:28604 | |
| Commercial assay, kit | MACHEREY-NAGEL plasmid extraction kit | MACHEREY-NAGEL | Ref:740588.250 | |
| Software, algorithm | Graphpad Prism 10 | Dotmatics | Version 10.4.0 (621) | |
| Software, algorithm | ImageJ-FIJI | ImageJ-FIJI | v1.54q | |
| Software, algorithm | Microsoft Excel | Microsoft | | |
| Software, algorithm | Microsoft Word | Microsoft | | |
| Software, algorithm | Adobe Illustrator | Adobe | SWITCH - ETH ZURICH | |

## Lead contact

Further information and requests for resources and reagents should be directed to and will be fulfilled by the Lead Contact, Yves Barral (yves.barral@bc.biol.ethz.ch).

## Materials availability

This study generated ~50 novel yeast strains, derived either by crossing or transformation. All strains, primers and RNA FISH probes are listed in *Supplementary file 1*. All resources are available to the public upon request. All the raw data used in this manuscript is available in *Source data 1*.

## Experimental model details

### Strains

All yeast strains and plasmids used in this study are listed in *Supplementary file 1*. Strains are derived from the S288C background. For every strain derived from a cross, an appropriate control strain was always selected from the same tetrad. Protein C-terminal yeGFP-tag and knock-out strains were generated using methods described in *Janke et al., 2004* All cultures were grown using standard conditions at 160 rpm, in YPD or synthetic drop-out medium supplemented with 0.1% BSA for aging chips (SD-medium; ForMedium, Norfolk, UK) at 25 °C or 30 °C.

### Intron-less strains

Intron deletions were conducted in two independent colonies of the strain JPY10I, as outlined previously in *Parenteau et al., 2008* Briefly, each intron was individually deleted using a modified two-step process (pop-in/pop-out), ensuring the absence of any additional sequence or markers post-deletion. The deletions were done in diploid strains, confirmed via PCR, and three distinct haploid strains bearing the confirmed deletions were chosen for subsequent analyses. Combinations of different Δ-intron strains have been derived by crossing and validated by PCR for the absence of the intron.

### Chromosome and plasmid reporter strains

Multiple chromosomes and minichromosome visualization strains have been used in this study: (1) Chromosome II reporter strain, with the centromere proximal TetO array (*POA1* locus) strain was kindly provided by Elisa Dultz from the Weis Lab, ETH, Institute of Biochemistry (internal reference KWY3759). (2) Chromosome IV TetO (centromere proximal) and LacO (centromere distal) were acquired from strains used in *Neurohr et al., 2011*; 17, originally derived from *Vas et al., 2007* (3) *Ubi-GFP minichromosome (PYB 2665) was derived by cloning UBIYdkGFP\* from Houser et al., 2012 (PNC 1136) Houser et al., 2012 into a PYB 2640 backbone by homologous recombination in yeast.* (4) *A minichromosome reporter containing the TetO array was previously described in Denoth Lippuner et al., 2014*.

### Plasmids

For the assessment of centromeric plasmid loss using the TeO-TetR-GFP system, we used the plasmid already published in *Denoth-Lippuner et al., 2014*. For the construction of a centromeric plasmid encoding GFP as a reporter, we used a short-lived GFP (UBIYΔkGFP\*–SpHis5–TIM9 in pUC118) available from Addgene(pNC1136). Ubi-GFP was amplified from PNC1136 with primers MM 311/312 (listed in *Supplementary file 1*). The resulting PCR product contained 50 bp homologous overhangs to another plasmid containing a centromere, pADH1 promoter and a pUG23 backbone (PYB2640). The template plasmid (PYB2640) was digested with MscI and Sac II and co-transformed with amplified UBIYΔkGFP\* PCR product to yeast to assemble the plasmid with *CEN* and pADH1-UBIYΔkGFP\* via homologous recombination. Successful recombinants were then selected on -*LEU* media after transformation. Recombinant plasmid was extracted from yeast and transformed into bacteria. The correct plasmid assembly was validated by sequencing. A full sequence of the assembled plasmid is available in the source data. For transformation of yeast with a non-centromeric plasmid, we have utilized Yeast Replicative Plasmid 17 (YRp17) from *Mann and Davis, 1983*. The full sequence of YRp17 is available online and in *Figure 1—source data 1*.

## Microscopy

For confocal fluorescent microscopy, yeast cells were precultured in YPD and then washed in synthetic drop-out medium. One ml of cells from exponentially growing cultures with OD <1 was concentrated by centrifugation at 1500 rcf, washed in SD complete, and resuspended in ~10 µl of low fluorescent SD-medium, spotted on a round coverslip and immobilized with a SD/agar patch. The cells were

imaged in >15 z-stack slices with 0.3 µm spacing, with a 100×/1.4 NA objective on a DeltaVision microscope (Applied Precision) equipped with a CCD HQ2 camera (Roper), 250 W Xenon lamps, Soft-worx software (Applied Precision), and a temperature chamber set to 30 °C.

## Aging microfluidic platform

Chromosome segregation and protein intensity during aging were assessed using the high-throughput yeast aging analysis (HYAA) microfluidics dissection platform (*Jo et al., 2015*). The PDMS (polydimethylsiloxane) microchannel is made by soft-lithography and bonded on the 30 mm micro-well cover glass in the 55 mm glass bottom dish (Cellvis, CA, USA). For lifespan analyses, a chip with a new cell trapping design was used (*Meinema et al., 2022*) to ensure excellent retention of old cells.

To start the aging experiment, yeast cells were pre-cultured for 24 hr in SD-full media supplemented with 0.1% Albumin Bovine Serum (protease-free BSA; Acros Organics, Geel, Belgium). Young cells from an exponentially growing culture were captured in the traps of the microfluidic chip; the chip was continuously flushed with fresh medium at a constant flow of 10 µl/min, using a Harvard PHD Ultra syringe pump (Harvard Apparatus, Holiston, MA, USA) with 60 ml syringe per lane, with inner diameter 26.7 mm (Becton Dickinson, Franklin Lakes, NJ, USA). Bright field images in a single z focal plane were recorded every 15 min throughout the duration of the entire experiment to measure replicative age of cells. To record fluorescent signals, images with a fluorescent lamp and at least 13 x z-stacks (0.5 µm step) were acquired in 12 hr intervals. For imaging, we used an epi-fluorescent microscope (TiE, Nikon Instruments, Tokyo, Japan) controlled by Micro-Manager 1.4.23 software (*Edelstein et al., 2014*), with a Plan Apo 60×1.4 NA objective. For fluorescence illumination of the GFP and mCherry-labeled proteins, a Lumencor Spectra-X LED Light Engine was used. Z stacks of >13 slices with 0.5 µm spacing were recorded for fluorescent imaging every 12 hr to cover the entire volume of the nucleus during the aging process. Transmitted light imaging was done in a single Z plane to minimize the light exposure of the cells. The age of the cell was defined by the number of daughter cells that emerged during the budding cycles (CBE).

## Chromosome loss assessment

For the assessment of chromosome and mini-chromosome loss and missegregation in aging, trapped cells in the chip that had the labeled chromosomes at the initial time point of 0 hr (~virtually all cells) were followed in transmitted light channel and fluorescent images were acquired every 12 hr. Only cells where the entire volume of the nucleus was in focus and imaged, and where SPB was clearly visible were considered in the analysis. The background fluorescence was subtracted in FIJI using a subtract background function. The aged nuclei were examined by scrolling through the z stack (>13 stacks, 0,5 µm step) in both channels to validate the presence of the SPBs and TetR foci in the mother or the daughter cell. 'Chromosome loss' refers to two quantified events: (1) The absence of the chromosome from a mother or a daughter cell compartment during anaphase (2) A G1 mother cell that just underwent its final division and is clearly viable, with a present SPB dot, but no chromosome dot. When 'Chromosome loss' is listed, it pertains to the sum of these two events. When 'chromosome loss in anaphase' is listed, it only pertains to the loss frequency observed during anaphase. Cells which lost chromosomes due to abnormal mitotic events in the last replicative division, such as SPB overamplification or full nuclei migration into the bud (Combined ~10% of all cells in the final replicative division) were discarded from the quantifications. All cells were examined in the DIC channel for signs of disrupted morphology indicating cell death and followed up in DIC to confirm cell death after chromosome loss. For the assessment of minichromosome loss in aging (TetO array or ubi-GFP minichromosome), the yeast cultures were grown in SD-LEU or SD-URA selective media enabling minichromosome maintenance in all starting young cells. When loaded onto an aging chip cell were transferred to SD complete media, allowing them to lose the minichromosome during aging without inducing cell death. As the cells were grown in selective media, the minichromosome loss rate at 0 hr is N/A. In general, aging chips are not suitable for measurement of chromosome loss frequency in young cells, as cells within the young population that are missegregating chromosomes are usually stressed and enlarged, and thus cannot be loaded into the microfluidic traps intended for normal young cells. For chromosome missegregation frequency in young cells, we have used classical confocal imaging on a patch of SD agar, as described in the Microscopy section.

## Measurements of chromosome loss per CBE

To measure chromosome loss probability per Completed Budding Event (CBE), we utilized at least 150 cells per genotype and quantified the number of cells that lost the reporter chromosome during a specific replicative division. The number of losses at a specific CBE was then divided by the total number of cells which passed through mitosis at that CBE number to give the probability of chromosome loss per CBE. Categories of 5 x CBE and its mean value of chromosome loss were plotted along with its SEM.

## Mother enrichment program (MEP) with FACS and RNA FISH

Mother Enrichment Program was performed according to the standard procedure, using the strains acquired from the Gotchling Lab (*Lindstrom and Gottschling, 2009*). Cells were labeled with biotin and aged for 28 hr, after which they were fixed in 4% PFA for 30 min. The cells were then stained with streptavidin conjugated with a FACS-compatible fluorophore (647 nm) and sorted using FACSAria III at the Flow Cytometry Core Facility at ETH Zurich to enrich for old mother cells. After FACS sorting, single molecule RNA FISH with Stellaris probes designed for introns of *GLC7, YRA1 and DBP2* was performed as described in *Rahman and Zenklusen, 2013*.

## Quantification and statistical analysis

Statistical analyses were performed using GraphPad Prism 10 software.

## Acknowledgements

We acknowledge Dr. Dan Jarosz, Dr. Gabriel Neurohr, Dr. Madhav Jagannathan, Dr. Patrick Meraldi, Dr. Anna Marzelliusardottir for critical reading and suggestions on the Manuscript. We acknowledge the ScopeM facility at the Institute of Biochemistry, ETH, for their technical assistance. We acknowledge Elisa Dultz for providing the Chromosome II strain with the POA1::TetO array. YB: grant 31003 A-105904 from the Swiss National Science Foundation (SNSF). MM: ETH postdoctoral fellowship 19–1 FEL-10. JP: CIHR (Canadian Institute of Health Research) 201809PJT-407549-BMB-CFDA-53473. SSL: Scope M facility of ETH Zurich, Institute of Biochemistry. SAE: CIHR (Canadian Institute of Health Research) 201809PJT-407549-BMB-CFDA-53473.

## Additional information

### Funding

| Funder | Grant reference number | Author |
|---|---|---|
| Schweizerischer Nationalfonds zur Förderung der Wissenschaftlichen Forschung | 31003A-105904 | Yves Barral |
| Canadian Institutes of Health Research | 201809PJT-407549-BMB-CFDA-53473 | Sherif Abou Elela |

The funders had no role in study design, data collection and interpretation, or the decision to submit the work for publication.

### Author contributions

Mihailo Mirkovic, Conceptualization, Data curation, Formal analysis, Validation, Investigation, Visualization, Methodology, Writing – original draft, Writing – review and editing; Jordan McCarthy, Formal analysis, Validation, Investigation, Writing – review and editing; Anne Cornelis Meinema, Conceptualization, Data curation, Investigation, Visualization; Julie Parenteau, Sung Sik Lee, Resources; Sherif Abou Elela, Conceptualization, Resources, Funding acquisition, Writing – review and editing; Yves Barral, Conceptualization, Supervision, Funding acquisition, Validation, Methodology, Writing – original draft, Project administration, Writing – review and editing

## Author ORCIDs
Mihailo Mirkovic (ID) https://orcid.org/0000-0003-0802-7200
Jordan McCarthy (ID) https://orcid.org/0000-0002-2824-4248
Anne Cornelis Meinema (ID) https://orcid.org/0000-0002-0002-3486
Julie Parenteau (ID) https://orcid.org/0000-0002-6007-7419
Sung Sik Lee (ID) https://orcid.org/0000-0001-9267-232X
Sherif Abou Elela (ID) https://orcid.org/0000-0002-0630-3294
Yves Barral (ID) https://orcid.org/0000-0002-0989-3373

Reviewer #1 (Public review): https://doi.org/10.7554/eLife.104530.3.sa1
Reviewer #2 (Public review): https://doi.org/10.7554/eLife.104530.3.sa2
Reviewer #3 (Public review): https://doi.org/10.7554/eLife.104530.3.sa3
Author response https://doi.org/10.7554/eLife.104530.3.sa4

## Additional files

### Supplementary files
Supplementary file 1. Strain genotypes, primer sequences, and single molecule RNA fluorescence in situ hybridization (smRNA FISH) probe sequences used in the study.

MDAR checklist

Source data 1. Raw data used to generate figures in this manuscript.

### Data availability
All the raw data in this manuscript is provided in the source data files.

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
