## [Editor Report · eLife Assessment]

This **fundamental** study reveals that aging in yeast leads to chromosome mis-segregation due to asymmetric partitioning of chromosomes, driven by disruption of the nuclear pore complex and pre-mRNA leakage. The findings are **convincingly** supported by carefully-designed experimental data with a combination of genetic, molecular biology and cell biology approaches.

---

## [Referee Report · Reviewer #1 (Public review)]

Summary:

In this study, the authors explore a novel mechanism linking aging to chromosome mis-segregation and aneuploidy in yeast cells. They reveal that, in old yeast mother cells, chromosome loss occurs through asymmetric partitioning of chromosomes to daughter cells, a process coupled with the inheritance of an old Spindle Pole Body. Remarkably, the authors identify that remodeling of the nuclear pore complex (NPC), specifically the displacement of its nuclear basket, triggers these asymmetric segregation events. This disruption also leads to the leakage of unspliced pre-mRNAs into the cytoplasm, highlighting a breakdown in RNA quality control. Through genetic manipulation, the study demonstrates that removing introns from key chromosome segregation genes is sufficient to prevent chromosome loss in aged cells. Moreover, promoting pre-mRNA leakage in young cells mimics the chromosome mis-segregation observed in old cells, providing further evidence for the critical role of nuclear envelope integrity and RNA processing in aging-related genome instability.

Strengths:

The findings presented are not only intriguing but also well-supported by robust experimental data, highlighting a previously unrecognized connection between nuclear envelope integrity, RNA processing, and genome stability in aging cells, deepening our understanding of the molecular basis of chromosome loss in aging.

Weaknesses:

The authors have satisfactorily addressed my concerns.

---

## [Referee Report · Reviewer #2 (Public review)]

Summary:

The authors make the interesting discovery of increased chromosome non-dysjunction in aging yeast mother cells. The phenotype is quite striking and well supported with solid experimental evidence. This is quite significant to a haploid cell (as used here) - loss of an essential chromosome leads to death soon thereafter. The authors then work to tie this phenotype to other age-associated phenotypes that have been previously characterized: accumulation of extrachromosomal rDNA circles that then correlate with compromised nuclear pore export functions, which correlates with "leaky" pores that permit unspliced mRNA messages to be inappropriately exported to the cytoplasm. They then infer that three intron containing mRNAs that encode portions in resolving sister chromatid separation during mitosis, are unspliced in this age-associated defect and thus lead to the non-dysjunction problem.

Strengths:

The discovery of age-associated chromosome non-dysjunction is an interesting discovery, and it is demonstrated in a convincing fashion with "classic" microscopy-based single cell fluorescent chromosome assays that are appropriate and seem robust. The correlation of this phenotype with other age-associated phenotypes - specifically extrachromosomal rDNA circles and nuclear pore dysfunction - is supported by in vivo genetic manipulations that have been well-characterized in the past.

In addition, the application of the single cell mRNA splicing defect reporter showed very convincingly that general mRNA splicing is compromised in aged cells. Such a pleiotropic event certainly has big implications.

Weaknesses:

The authors have addressed my major concerns with experimentation or clarification.

---

## [Referee Report · Reviewer #3 (Public review)]

Summary:

Mirkovic et al explore the cause underlying development of aneuploidy during aging. This paper provides a compelling insight into the basis of chromosome missegregation in aged cells, tying this phenomenon to the established Nuclear Pore Complex architecture remodeling that occurs with aging across a large span of diverse organisms. The authors first establish that aged mother cells exhibit aberrant error correction during mitosis. As extrachromosomal rDNA circles (ERCs) are known to increase with age and lead to NPC dysfunction that can result in leakage of unspliced pre-mRNAs, Mirkovic et al search for intron-containing genes in yeast that may be underlying chromosome missegregation, identifying three genes in the aurora B-dependent error correction pathway: MCM21, NBL1, and GLC7. Interestingly, intron-less mutants in these genes suppress chromosome loss in aged cells, with a significant impact observed when all three introns were deleted (3x∆i). The 3x∆i mutant also suppresses the increased chromosome loss resulting from nuclear basket destabilization in a mlp1∆ mutant. The authors then directly test if aged cells do exhibit aberrant mRNA export, using RNA FISH to identify that old cells indeed leak intron-containing pre-mRNA into the cytoplasm, as well as a reporter assay to demonstrate translation of leaked pre-mRNA, and that this is suppressed in cells producing less ERCs. Mutants causing increased pre-mRNA leakage are sufficient to induce chromosome missegregation, which is suppressed by the 3x∆i.

Strengths:

The finding that deleting the introns of 3 genes in the Aurora B pathway can suppress age-related chromosome missegregation is highly compelling. Additionally, the rationale behind the various experiments in this paper is well-reasoned and clearly explained.

Weaknesses:

My main concerns have been thoroughly addressed by the authors.

---

## [Author Response]

The following is the authors’ response to the original reviews.

**Reviewer #1 (Public review):**
Summary:In this study, the authors explore a novel mechanism linking aging to chromosome mis-segregation and aneuploidy in yeast cells. They reveal that, in old yeast mother cells, chromosome loss occurs through asymmetric partitioning of chromosomes to daughter cells, a process coupled with the inheritance of an old Spindle Pole Body. Remarkably, the authors identify that remodelling of the nuclear pore complex (NPC), specifically the displacement of its nuclear basket, triggers these asymmetric segregation events. This disruption also leads to the leakage of unspliced pre-mRNAs into the cytoplasm, highlighting a breakdown in RNA quality control. Through genetic manipulation, the study demonstrates that removing introns from key chromosome segregation genes is sufficient to prevent chromosome loss in aged cells. Moreover, promoting pre-mRNA leakage in young cells mimics the chromosome mis-segregation observed in old cells, providing further evidence for the critical role of nuclear envelope integrity and RNA processing in aging-related genome instability.Strengths:The findings presented are not only intriguing but also well-supported by robust experimental data, highlighting a previously unrecognized connection between nuclear envelope integrity, RNA processing, and genome stability in aging cells, deepening our understanding of the molecular basis of chromosome loss in aging.

We thank the reviewer for this very positive assessment of our work

Weaknesses:Further analysis of yeast aging data from microfluidic experiments will provide important information about the dynamic features and prevalence of the key aging phenotypes, e.g. pre-mRNA leakage and chromosome loss, reported in this work.

We thank the reviewer for bringing this point, which we have addressed in the revised version of the manuscript. In short, chromosome loss is an abrupt, late event in the lifespan of the cells. To examine its prevalence, we have quantified the combined loss frequency of two chromosomes when both are labelled in the same cell. Whereas single chromosomes are lost at a frequency of 10-15% per cell, less than 5% of the cells lose both at the same time. Thus, the different chromosomes are lost largely but not fully independently from each other. Based on these data, and on the fact that yeast cells have 16 chromosomes, we evaluate that about half of the cells lose at least one chromosome in their final cell cycle.

We also tried to estimate the prevalence of the pre-mRNA leakage phenotype, based on the increased mCherry to GFP ratio observed between 0h and 24 hours of aging for 146 individual cells. For this analysis, we compared the mCherry/GFP ratio at 0 and 24h for the same individual cell. This analysis indicates that 81% of the cells show a fold change strictly above 1 as they age. Furthermore, the data appears to be unimodal. Thus, we can conservatively conclude that a majority of the cells show premRNA leakage at 24 hours. Since not all cells are at the end of their life at that time, this is possibly an underestimate.

In addition, a discussion would be needed to clarify the relationship between "chromosome loss" in this study and "genomic missegregation" reported previously in yeast aging.

Genomic mis-segregation is characterized by the entry of both SPBs and all the chromosomes into the daughter cell compartment (PMID: 31714209). We have observed these events in our movies as well. However, the chromosome loss phenotype that we are focusing on affects only some chromosomes (as discussed above) and takes place under proper elongation of the spindle, with one SPB remaining in the mother cell whereas the other one goes to the bud, as shown in the manuscript’s Figure 2. In our movies, chromosome loss is at least three-fold more frequent (for a single chromosome) than full genome mis-segregation (Sup Fig 1A-B). Furthermore, whereas chromosome loss is alleviated by the removal of the introns of MCM21, NBL1 and GLC7, genomic mis-segregation is not (Sup Fig 1B). Thus, genomic mis-segregation mentioned by the reviewer is a process distinct from the chromosome loss that we report. This discussion and the relevant data have been added to the manuscript.

We thank the reviewer for bringing up the possible confusion between these two phenotypes, allowing us to clarify this point.

**Reviewer #2 (Public review):**
Summary:The authors make the interesting discovery of increased chromosome non-dysjunction in aging yeast mother cells. The phenotype is quite striking and well supported with solid experimental evidence. This is quite significant to a haploid cell (as used here) - loss of an essential chromosome leads to death soon thereafter. The authors then work to tie this phenotype to other age-associated phenotypes that have been previously characterized: accumulation of extrachromosomal rDNA circles that then correlate with compromised nuclear pore export functions, which correlates with "leaky" pores that permit unspliced mRNA messages to be inappropriately exported to the cytoplasm. They then infer that three intron containing mRNAs that encode portions in resolving sister chromatid separation during mitosis, are unspliced in this age-associated defect and thus lead to the non-dysjunction problem.Strengths: The discovery of age-associated chromosome non-dysjunction is an interesting discovery, and it is demonstrated in a convincing fashion with "classic" microscopy-based single cell fluorescent chromosome assays that are appropriate and seem robust. The correlation of this phenotype with other age-associated phenotypes - specifically extrachromosomal rDNA circles and nuclear pore dysfunction - is supported by in vivo genetic manipulations that have been well-characterized in the past.In addition, the application of the single cell mRNA splicing defect reporter showed very convincingly that general mRNA splicing is compromised in aged cells. Such a pleiotropic event certainly has big implications.

We thank the reviewer for this assessment of our work. To avoid confusion, we would like to stress out, however, that our data do not show that splicing per se is defective in old cells. Actually, we specifically show that the cells are unlikely to show splicing defect (last figure of the original and the revised version of the manuscript). Our data specifically show that unspliced mRNAs tend to leak out of the nucleus of old cells.

Weaknesses:The biggest weakness is "connecting all the dots" of causality and linking the splicing defect to chromosome disjunction. I commend the authors for making a valiant effort in this regard, but there are many caveats to this interpretation. While the "triple intron" removal suppressed the non-dysjunction defect in aged cells, this could simply be a kinetic fix, where a slowdown in the relevant aspects of mitosis, could give the cell time to resolve the syntelic attachment of the chromatids.

The possibility that intron-removal leads to a kinetic fix is an interesting idea that we have now considered. In the revised manuscript, we now provide measurements of mitotic duration in the “triple intron” mutant compared to wild type cells and the duration of their last cell cycle (See supplementary figure 3A-D). There is no evidence that removing these introns slows down mitosis. Thus, the kinetic fix hypothesis is unlikely to explain our observation about the effect of intron removal.

To this point, I note that the intron-less version of GLC7, which affects the most dramatic suppression of the three genes, is reported by one of the authors to have a slow growth rate (Parenteau et al, 2008 - https://doi.org/10.1091/mbc.e07-12-1254)

The reviewer is right, removing the intron of GLC7 reduces the expression levels of the gene product (PMID: 16816425) to about 50% of the original value and causes a slow growth phenotype. However, the cells revert fairly rapidly through duplication of the GLC7-∆i gene (see supplementary Figure 3EF). As a consequence, neither the GLC7-∆i nor the 3x∆i mutant strains show noticeable growth phenotypes by spot assays. We now document these findings in supplementary figure 3.

Lastly, the Herculean effort to perform FISH of the introns in the cytoplasm is quite literally at the statistical limit of this assay. The data were not as robust as the other assays employed through this study. The data show either "no" signal for the young cells or a signal of 0, 1, or 2 FISH foci in the aged cells. In a Poisson distribution, which this follows, it is improbable to distinguish between these differences.

This is correct, this experiment was not the easiest of the manuscript... However, despite the limitations of the assay, the data presented in figure 7B are very clear. 300 cells aged by MEP were analysed, divided in the cohorts of 100 each, and the distribution of foci (nuclear vs cytoplasmic) in these aged cells were compared to the distribution in three cohorts of young cells. For all 3 aged cohorts, over 70% of the visible foci were cytoplasmic, while in the young cells, this figure was around 3%. A t-test was conducted to compare these frequencies between young and old cells (Figure 7B). The difference is highly significant. Therefore, we are clearly not at the statistical limit.

What the reviewer refers to is the supplementary Figure 4, where we were simply asking (i) is the signal lost in cells lacking the intron of GLC7 (the response is unambiguously yes) and (ii) what is the general number of dots per cell between young and old wild type cells (without distinguishing between nuclear and cytoplasmic) and the information to be taken from this last quantification is indeed that there is no clearly distinguishable difference between these two population of cells, as the reviewer rightly concludes. In other word, the reason why there are more dots in the cytoplasm of the old cells in the Figure 7B is not because the old cells have much more dots in general (see supplementary Figure 4C). We hope that these clarifications help understand the data better. We have edited the manuscript to avoid confusion.

**Reviewer #3 (Public review):**
Summary:Mirkovic et al explore the cause underlying development of aneuploidy during aging. This paper provides a compelling insight into the basis of chromosome missegregation in aged cells, tying this phenomenon to the established Nuclear Pore Complex architecture remodelling that occurs with aging across a large span of diverse organisms. The authors first establish that aged mother cells exhibit aberrant error correction during mitosis. As extrachromosomal rDNA circles (ERCs) are known to increase with age and lead to NPC dysfunction that can result in leakage of unspliced pre-mRNAs, Mirkovic et al search for intron-containing genes in yeast that may be underlying chromosome missegregation, identifying three genes in the aurora B-dependent error correction pathway: MCM21, NBL1, and GLC7. Interestingly, intron-less mutants in these genes suppress chromosome loss in aged cells, with a significant impact observed when all three introns were deleted (3x∆i). The 3x∆i mutant also suppresses the increased chromosome loss resulting from nuclear basket destabilization in a mlp1∆ mutant. The authors then directly test if aged cells do exhibit aberrant mRNA export, using RNA FISH to identify that old cells indeed leak intron-containing pre-mRNA into the cytoplasm, as well as a reporter assay to demonstrate translation of leaked pre-mRNA, and that this is suppressed in cells producing less ERCs. Mutants causing increased pre-mRNA leakage are sufficient to induce chromosome missegregation, which is suppressed by the 3x∆i.Strengths:The finding that deleting the introns of 3 genes in the Aurora B pathway can suppress age-related chromosome missegregation is highly compelling. Additionally, the rationale behind the various experiments in this paper is well-reasoned and clearly explained.

We thank the reviewer for their very positive assessment of our work

Weaknesses:In some cases, controls for experiments were not presented or were depicted in other figures.

We are sorry about this confusion. We have improved our presentation of the controls, bringing them back each time they are relevant. We have also added those that were missing (such as those mentioned by reviewer 2, see above). Note that the frequencies of centromeric plasmid loss at 0h in Figure 1C is not meaningful and therefore not presented. Since the cells were grown on selective medium before loading on to the ageing chip, we cannot report a plasmid loss frequency here. The ageing experiments themselves were subsequently conducted in full medium, to allow for centromeric plasmid loss without killing the cell. We explain this in the materials and methods section.

High variability was seen in chromosome loss data, leading to large error bars.

We thank the reviewer for this comment. The variance in those two figures (3A and 5D) comes from the suboptimal plotting of this data. This is now corrected as follows. We divided the available data into 4 cohorts and then plotted the average loss frequency across these cohorts for the indicated age groups. This filters out much of the noise and improves the statistical resolution.

The text could have been more polished.

Thank you for this comment. We have gone through the manuscript again in detail.

**Reviewer #1 (Recommendations for the authors)**:(1) A previous study (PMID: 31714209). showed that aging yeast cells undergo genomic missegregation in which material was abnormally segregated to the daughter cells, leading to cell cycle arrest. After that, the missegregation is either corrected by returning aberrantly segregated genetic material to the mother cells so that they can resume cell cycles, or if not corrected, the mother cells will terminally exist the cell cycle and eventually die. That paper also showed that this agedependent genomic missegregation is related to rDNA instability. Is the chromosome loss in this work related to the genomic missegregation reported before? Is it partially reversible like genomic missegregation? Are all the chromosomes lost in one cell division, like in the case of genomic missegregation? Some additional characterization and a discussion would be helpful.

As mentioned above, indeed the phenotype of full genome mis-segregation described by Crane et al. (2019) is observable in our data as well. At 24h ~3% of the cells segregate both SPBs to the bud, as they previously described (Supp Figure 1A and B). This phenomenon is clearly distinct from asymmetric chromosome partition, where cells undergo anaphase, separate the SPBs and segregate one to the mother cell and one to the bud (Figure 2A). Also, asymmetric chromosome partitioning affects only a subset of the chromosomes (see below), not the entire genome. Finally, unlike asymmetric chromosome partitioning, the frequency of genome mis-segregation in ageing was not alleviated by intron removal (Supp Figure 1B). Thus, these two processes are clearly distinct and driven by different mechanisms. Note that asymmetric chromosome partitioning appears 3 to 5 times more frequently than genomic mis-segregation.

Supporting further the notion that these two processes are distinct, chromosome loss seals the end of the life of the cell, as we reported, indicating that this is not a reversible event. Also, it does not involve all chromosomes at once. Cells that contain the labelled versions of both chromosome II and IV at the same time, the loss frequency of both chromosomes is less than 5%, whereas each chromosome is lost in 10-15% of the cells (Figure 1C). Thus, most cells lose one and keep the other. Furthermore, this indicates that there are many more cells losing at least one chromosome than the 15% that lose chromosome IV for example, probably 50% or more. Thus, chromosome loss by asymmetric segregation is much more frequent than the partly transient transfer of the entire nucleus to the bud.

(2) What percentage of aging WT cells undergo pre-mRNA leakage (using the GFP/mCherry reporter) during their entire lifespan? Is it a sporadic, reversible process or an accumulative, one-way deterioration? Previous studies (PMID: 32675375; PMID: 24332850; PMID: 36194205; PMID: 31291577) showed that only a fraction of yeast cells age with rDNA instability and ERC accumulation, as indicated by excessive rRNA transcription and nucleolar enlargement. Are they the same fraction of aging cells that undergo pre-mRNA leakage and chromosome loss? This information will indicate the prevalence of the key aging phenotypes reported in this work and should be readily obtainable from microfluidic experiments. In addition, a careful discussion would be helpful.

Pre-mRNA leakage is relatively widespread in the population, but it is difficult to put a precise number on it. Analysis of how the mCherry/GFP ratio changes in 146 individual cells between 0 and 24 hours and imaging in our microfluidics platform indicates that ~80% show an increase and 50% of the cells show an increase above 1.5-fold. Therefore, the frequencies of pre-mRNA leakage and chromosome loss are probably similar. We have modified the discussion to account for these considerations. This would be in the same range as the frequency of aging by ERC accumulation (mode 1) estimated by PMID: 32675375.

**Reviewer #2 (Recommendations for the authors)**
The manuscript could use a bit of editing in places - please go through it once more.Editing suggestions:Line 80 – irrespective

Corrected.

Line 97 - these are not "rates" but frequencies. Please correct this error throughout.Replaced “rate” with “frequency throughout the manuscript and the figures, when pertaining to chromosome lossLine 328 - increase in chromosome...

Corrected.

Line 379 - tampering**Reviewer #3(Recommendations for the authors)**:Specific Feedback to Authors(a) Major Points(i) While the proposed connection between ERC-mediated nuclear basket removal and erroneous error correction was clearly stated, this connection is correlative and was not directly tested. Specifically, although mutants impacting ERC levels were tested for missegregation, it was not directly tested if increased missegregation levels occurred due to ERC tethering to the NPC and subsequent nuclear basket removal. It is possible that the increased ERCs may be driving missegregation via a different pathway. Authors should consider experiments to strengthen this idea, such as looking at chromosome loss frequency in a sir2∆ 3x∆i double mutant, or a sir2∆ sgf73∆ double mutant.

This connection is addressed in the original version of the manuscript, where we show that preventing attachment of ERCs to the NPC, by removing the linker protein Sgf73, alleviates chromosome loss. The link is further substantiated by the fact that removing the basket on its own promote chromosome loss and that in both cases, namely during normal aging, i.e., upon ERC accumulation, and upon basket removal the mechanism of chromosome loss is the same. In both cases, it depends on the introns of the GLC7, MCM21 and NBL1 genes.

However, we acknowledge that the mutants tested have pleiotropic effects, making interpretation somewhat difficult, even when examining chromosome loss in multiple mutants that affect ERC formation and NPC remodelling, as we have done. As recommended by the reviewer, we have characterized the phenotype of the sir2∆ 3x∆i mutant strain. Intron removal in the sir2∆ mutant cells largely rescued the elevated chromosome loss frequency of these cells and slightly extended their replicative lifespan (Figure 6D-E). We conclude that intron removal can remedy the chromosome loss phenotype of the sir2∆. Although clearly significant, the effect on the replicative lifespan was not very strong, likely due to the sir2∆ affecting other ageing processes.

Touching on this question, we added a new set of experiments asking whether any accumulating DNA circle causes chromosome loss in an intron-dependent manner. Thus, we have introduced a noncentromeric replicative plasmid in wild type and 3x∆i mutant strains carrying the labelled version of chromosome II (Figure 6A-C). These studies show that these cells age much faster than wild type cells, as expected, and lose chromosomes at a higher frequency than non-transformed cells. Finally, the effect is at least in part alleviated by removing the introns of NBL1, MCM21 and GLC7.

Therefore, after adding this new and more direct test of the role of DNA circles in chromosome loss, we are confidently concluding that ERC-mediated basket removal is the trigger of chromosome loss in old cells.

(b) Minor Points(i) In Figure 1C, the text (lines 91-92) argues that chromosome loss happens abruptly as cells age; however the data only show loss at young and old time points, not an intermediate, which leaves open the possibility that chromosome loss is occurring gradually. While cells that lost chromosomes should fail to divide further, we don't know if these events happened and were simply excluded.

We agree with the reviewer that formally the conclusion drawn in the lines 91-92 (of the original manuscript), namely that chromosome loss takes place abruptly as cells age, cannot be drawn from the Figure 1C alone but only from subsequent observations. However, since chromosome loss is lethal in haploid, as we mention in the text and the reviewer notes as well, it is difficult to envision how cells could lose chromosomes before the end of their lifespan and must therefore increase abruptly as the cells reach that point. This is now underlined in the revised version of the manuscript. Accordingly, the frequency of chromosome loss per age group, which is depicted in Figure 3A, shows that the wild type cells that have budded less than 10 times show no chromosome loss. The chromosome loss frequency starts to ramp up only pass that point. Therefore, chromosome loss does not increase linearly with age.

Additionally, cells that lost minichromosome should not arrest. We suggest that the interpretation of these data should be softened in the text, or that chromosome loss fraction could be more effectively portrayed as a Kaplan-Meier survival curve depicting cells that have not lost chromosomes, if these data are easily available. Or, chromosome loss at an intermediate time point could be depicted.

Since we cannot visualize more than 2 chromosomes at a time, it is not possible to plot the KaplanMeier curve of cells that have not lost chromosomes. However, as mentioned above, the chromosome loss frequencies at intermediate time points are depicted in Figure 3A and Figure 4B and shows that it increases with age.

(ii) Also regarding Figure 1, it would be helpful to expound on the purpose of the minichromosomes, as well as how the Ubi-GFP minichromosome is constructed.

We now explained why we tested the loss of minichromosome, namely, as a mean to test whether the centromere is necessary and sufficient to drive the loss of the genetic material linked to it, i.e., chromosomes, in old cells. Concerning the Ubi-GFP minichromosome, the Materials and methods section is now updated and reports plasmid construction, backbone used, primers as well as the plasmid sequence being available in the supplementary data.

The purpose of the minichromosome initially appears to be the engineering of an eccDNA (ERC) with a CEN to demonstrate distinct behaviour, but it is unclear whether this was actually conducted or if the minichromosome are simply CEN plasmids and/or if this was the intended goal. Furthermore, lines 102-103 state that the presence of a centromere was necessary and sufficient for minichromosome loss. However, since no constructs lacking a centromere were tested, necessity cannot be concluded. Please clarify this in the text and include experimental details to help readers understand what was tested.

We apologize for having been too short here. The behaviour of the CEN-less version of this plasmid has been characterized in detail in previous studies (Shcheprova et al., 2008; Denoth-Lippuner 2014, Meinema et al 2022). Here we focused on the behaviour of the CEN+ version of an otherwise Identical plasmid. We now clarify in the text that this plasmid is retained in the mother cell when CEN-less and cite the relevant literature.

(iii) It is unclear how cells at 0-3 budding events were identified in assays using the microfluidics platform. Can the authors clarify the known "age" of the cells once captured, i.e. how do the authors know how many divisions a cell has undergone prior to capture?

The reviewer is right; we do not know the exact age of these cells. However, in any asynchronous population of yeast cells, which is what we start from, 50% of the cells are newborn daughters, 25% have budded once, 12.5 have budded twice, 6.25 % have budded three times… Therefore, at the time of loading, 93% of the cells have budded between 0 and 3 times. For this reason, we report to this population as cells age 0-3 CBE. We acknowledge that this is an approximation, but it remains a relatively safe one.

(iv) While the schematic in Figure 2D is generally helpful, a different depiction of the old and new SPBs would be beneficial in cases where the new SPB and TetR-GFP are depicted as colocalized, it is difficult to see that the red is fainter for the new SPB.

We have corrected this issue by completely separating the SPB and the Chromosome signals in the Figure 2D.

(v) In Figure 2F, the grey colour of the 12h Ipl1-321 data bar did not have high enough contrast when the manuscript was printed-would recommend changing this to a darker shade.

We have corrected this issue by using a darker shade of grey.

(vi) In Figure 3A, 'Budding' is misspelled on X-axis label

We have corrected this error.

(vii) In Figure 4, the authors should clarify the differences between the analyses in panels B and C. The distinction is not immediately clear and may be difficult to grasp upon initial reading.

We have corrected this issue in the main text as well as figure legend.

(viii) In Figure 5, It would aid comparisons to depict the 3x∆i only as well on panels B, D, and E.

We have added 3x∆i data to Figure 5,6 and 8.

(ix) In Figure 6D, it is unclear why there was an appreciable level of unspliced RNA in the wild-type and sir2∆ young cells. Additionally, it is unclear why there is so much signal observed in the Merge image for the old wild-type cell, especially regarding the apparent bright spot. Is that nuclear signal? Please clarify.

The pre-mRNA processing reporter is not very efficiently spliced. It was selected as such during design (Sorenson et al 2014; DOI: 10.1261/rna.042663.113) to provide sensitivity. As for the bright spot occurring, translation of the unspliced reporter produces the N-terminal part of a ribosomal protein, a fraction of which forms some sort of nuclear aggregate in a fraction of the population.

(x) In Figure 6E, why does the sir2∆ exhibit higher mCherry/GFP than the wild-type and fob1∆ at "young age"? Is this due to disrupted proteostasis in the sir2∆, or a different pleiotropic effect of sir2∆? Please comment on this observation in the text.

Indeed, as we have stated in the text the sir2∆ mutation already perturbs pre-mRNA processing in young cells. We do not know the reason of this but indeed it is most probably reflective of its pleiotropic function. Following the reviewer’s request, we now state this in the text. For example, Sir2 may regulate the acetylation state of the basket itself. The genetic interactions observed between sir2∆ and quite a few nucleoporin mutations seem to support this possibility.

(xi) Throughout, the authors switch between depicting aging in Completed Budding Events versus hours, which made it difficult to compare data across figures

Ideally, all the data in this manuscript should be plotted according to the CBE age of the cell. To ensure that the major findings are plotted in such a way, we have done so for over ~3000 combined cells and thousands of replicative divisions in Figures 3,5-7. All the measurements of chromosome loss at a specific CBE had to be done manually, due to the absence of algorithms that would be able to accurately detect chromosome loss and replicative age. Therefore, doing this for the entirety of our dataset, encompassing well over 50 ageing chips and tens of thousands of cells is not easily doable at this stage.

(xii) Typo on line 12 (Sindle Pole Body)

We have corrected this error.

(xiii) The phrase should be 'chromosome partitioning' rather than 'chromosome partition', throughoutfor example, line 17

Replaced “chromosome partition” with “chromosome partitioning” throughout the text.

(xiv) There are inconsistencies between plural and singular references throughout sentences-example, lines 35-37, and lines 44-45.

We carefully combed through the manuscript again and hope that we caught all inconsistencies.